# Structural Inference of Dynamical Systems with Conjoined State Space Models

**Aoran Wang** [1]  &  **Jun Pang** [1,2]
[1] Faculty of Science, Technology and Medicine, University of Luxembourg
[2] Institute for Advanced Studies, University of Luxembourg
{aoran.wang, jun.pang}@uni.lu

## Abstract

This paper introduces SICSM, a novel structural inference framework that integrates Selective State Space Models (selective SSMs) with Generative Flow Networks (GFNs) to handle the challenges posed by dynamical systems with irregularly sampled trajectories and partial observations. By utilizing the robust temporal modeling capabilities of selective SSMs, our approach learns input-dependent transition functions that adapt to non-uniform time intervals, thereby enhancing the accuracy of structural inference. By aggregating dynamics across diverse temporal dependencies and channeling them into the GFN, the SICSM adeptly approximates the posterior distribution of the system's structure. This process not only enables precise inference of complex interactions within partially observed systems but also ensures the seamless integration of prior knowledge, enhancing the model's accuracy and robustness. Extensive evaluations on sixteen diverse datasets demonstrate that SICSM outperforms existing methods, particularly in scenarios characterized by irregular sampling and incomplete observations, which highlight its potential as a reliable tool for scientific discovery and system diagnostics in disciplines that demand precise modeling of complex interactions.

## 1 Introduction

In the complex real-world phenomena, many dynamical systems manifest as networks of interacting entities. These systems are effectively modeled as graphs where nodes represent the agents, edges depict the interactions, and the adjacency matrix captures the structural essence of these interactions. Such representations are crucial across various domains, from intricate physical systems [32, 23, 59] and multi-agent systems [10, 34], to complex biological architectures [49, 44]. Unveiling the hidden structures within these networks is not only academically enriching but also essential for enhancing our understanding of the systems' intrinsic mechanisms and improving our ability to predict and manage their behaviors. However, this task becomes challenging when the observable data, often limited to features of agents within specific time frames, conceals the underlying structural dynamics. This limitation necessitates robust *structural inference* methodologies capable of discerning the latent structures from the *trajectories*—the observable features of all agents over a given period.

As the field of scientific discovery advances, particularly with the integration of neural network technologies, structural inference has emerged as a key method for decoding the complex interactions within dynamical systems from trajectories [31, 2, 57, 12, 39, 51, 16, 54, 64]. This process is essential for understanding and predicting system behaviors but encounters significant challenges, particularly when addressing irregularly sampled data and partially observed some nodes of a dynamical system, which are typified by unequal sampling time intervals and the presence of unobserved nodes. Conventional methods, including those based on Variational Autoencoders (VAEs) [30], have pioneered some paths but often struggle with datasets characterized by non-

uniform sampling rates and incomplete observations. These methods typically require uniform data and have difficulty managing indirect or obscured node interactions [31, 2, 57, 12, 39, 51, 54], highlighting a critical need for more adaptable and resilient inference models.

To overcome these limitations, this paper introduces a novel framework, Structural Inference with Conjoined State Space Models (SICSM), which conjoins Selective State Space Models (selective SSMs) [20] with a Generative Flow Network (GFN) [7, 8]. This innovative approach leverages the robust temporal modeling capabilities of selective SSMs alongside the flexible, data-driven structural inference provided by the GFN. SICSM is specifically designed to address the challenges of irregular sampling and partial observability, enhancing inference accuracy and robustness through the sophisticated integration of prior knowledge and adaptive learning mechanisms. Central to SICSM is its ability to learn input-dependent transition functions that dynamically adjust to the timing of data points, crucial for managing datasets with irregular intervals. Moreover, by aggregating outputs from multiple Residual Blocks containing an selective SSM in each, SICSM offers a rich representation of system dynamics, enabling more precise reconstruction of node interactions within partially observed systems. Our comprehensive evaluations across a variety of datasets—from mechanical systems like spring simulations to biological networks depicting gene expressions—demonstrate SICSM's superior performance over existing methods. Its robustness shines particularly in its ability to maintain high structural inference accuracy under diverse and challenging conditions, affirming its potential as an essential tool for scientific discovery and system diagnostics in disciplines that demand intricate, accurate modeling of complex systems. In essence, SICSM not only redefines approaches to structural inference in complex systems but also paves new research avenues previously limited by data sampling and observability constraints. With further refinement, this approach is poised to transform our understanding of interactions with dynamical systems across multiple scientific fields. Our contributions encompass the following aspects:

- We develop a novel framework, SICSM, that integrates Selective State Space Models with a Generative Flow Network to enhance structural inference.
- We introduce adaptive mechanisms within SICSM that effectively handle irregular sampling and partial observability, significantly enhancing the model's applicability to real-world datasets.
- SICSM employs a novel approach by aggregating outputs from multiple Residual Blocks, which enables it to capture a deeper and more detailed representation of dynamic system interactions.
- Extensive validation demonstrates its superiority over baselines in reconstructing complex structures, especially under challenging conditions of irregular sampling and incomplete observations.

## 2 Related Work

**Structural Inference.** Structural inference aims to uncover the hidden structure of complex systems using observed trajectories. A pivotal contribution in this area is Neural Relational Inference (NRI) [31], which leverages a VAE within a fixed, fully connected graph framework. Building on NRI, subsequent research has expanded the domain of structural inference. Recent advancements include handling multi-interaction systems [57], integrating efficient message-passing [12], incorporating modular meta-learning [2], iteratively pruning indirect connections [51], developing structural inference with reservoir computing [54], and applying deep active learning to complex systems [52]. Other techniques involve reconstructing trajectories by minimizing relation potentials [16], computing partial correlation coefficients based on node embeddings [53], and with diffusion process [64]. Existing methods frequently employ the Variational Information Bottleneck principle [1], which often requires regularly sampled trajectories and complete observation of all nodes. Incorporating known interactions into VAEs necessitates complex adjustments, combining unsupervised and supervised learning in the latent space. This complexity complicates generalization to unlabeled edges.

**State Space Models.** State space models (SSMs) update sequences through recurrent hidden states. The selective state space architecture known as Mamba [20], recently emerged as an efficient and flexible design, using recurrent scans and a selection mechanism to control sequence flow into hidden states. Mamba shows promise across time-series tasks [56, 38, 43], video analysis [35, 61, 37] and healthcare applications [40, 45, 60]. In addition, several studies explore graph modeling with Mamba [55, 36, 6], but none of them apply it for structural inference from observational trajectories. In this work, we employ the fundamental operating mechanism, the selective SSM module, in the form of stacking blocks, to model observational trajectories and to deal with the challenge of irregular sampling as well as partial observation.

**Generative Flow Networks.** Generative flow networks (GFNs) excel in generating and sampling discrete states from high-dimensional distributions [7, 8]. Recent research has explored topics such as amortized inference [33], Bayesian structure learning [17], combinatorial optimization [63], biological sequence design [27], and broader scientific discovery [28]. Work in network inference [17, 18, 4] focuses on Bayesian inference to maintain system structure within state spaces. Our approach leverages Residual Blocks to learn one-dimensional embeddings from multi-dimensional features, enabling GFN to effectively learn while preserving dynamic complexity within the embeddings.

## 3 Preliminaries

### 3.1 Notations and Problem Formulation

We model a dynamical system as a directed graph $G = (\mathcal{V}, E)$, representing agents as nodes and interactions as directed edges. The graph consists of a node feature set $\mathcal{V}$ with $n$ nodes and an edge set $E$. Node features evolve over time, forming trajectories $\mathcal{V} = \{V\} = \{V^0, V^1, \ldots, V^{T-1}\}$ across $T$ time steps, where $V^t$ represents the feature set at time $t$. Each node feature $v_i^t \in \mathbb{R}^d$ is $d$-dimensional. For irregularly sampled trajectories, the $T$ time steps may have different intervals. And for partial observation of the dynamical system, we expect the count of observed nodes $n$ is smaller than the total count of nodes $n_{tot}$ in the system. We observe a set of $M$ trajectories: $\{V_{[1]}, V_{[2]}, \ldots, V_{[M]}\}$, assuming a static edge set $E$. An asymmetric adjacency matrix $\mathbf{Adj}$ is derived from $E$, where $\mathbf{Adj}_{ij} \in \{0, 1\}$ indicates the presence or absence of a directed edge. In dynamical systems, the dynamics of node $i$ at time $t + 1$ are influenced by $\mathbf{Adj}$ as follows:

$$v_i^{t+1} = v_i^t + \Delta \cdot \sum_{j \in \mathcal{U}_i} f(||v_i, v_j||_\alpha), \tag{1}$$

where $\Delta$ is the time interval, $\mathcal{U}_i$ denotes the set of nodes connected to node $i$ which is derived from $\mathbf{Adj}$, $f(\cdot)$ represents the state-transition function, and $||\cdot, \cdot||_\alpha$ is the $\alpha$-distance.

For an illustrative example, we may consider a dynamical system comprising $n = 10$ balls connected by springs, representing $n = 10$ nodes $\mathcal{V}$ and directed edges $E$, respectively. Initially, we set the positions and velocities of each ball, so that each node feature $v_i^t \in \mathbb{R}^d$ is $d$-dimensional where $d = 4$ in this example. We then let them move under the influence of spring forces, which arise from the structural connections (edges) between the balls (nodes). Over the observation period, these balls change their positions and velocities. And we record the trajectories as the collection of the evolving features of all nodes: $\mathcal{V} = \{V\} = \{V^0, V^1, \ldots, V^{T-1}\}$ across $T$ time steps, where $V^t$ represents the feature set at time $t$. In total we observe a set of $M$ trajectories: $\{V_{[1]}, V_{[2]}, \ldots, V_{[M]}\}$, assuming a static edge set $E$. Suppose we initially lack knowledge of which balls are connected, i.e., $E$ is unknown; the task of structural inference in this scenario would involve deducing the connectivity between the balls based on their observed trajectories, represented by either the edge set $E$ or the adjacency matrix $\mathbf{Adj}$.

In this work, we introduce SICSM, a novel structural inference method with conjoined state modeling, designed to handle both irregularly sampled trajectories and systems with partial observations.

### 3.2 State Space Models

SSMs capture the behavior of dynamical systems by modeling the internal state and relationships between latent states $h^t \in \mathbb{R}^N$, input sequences $x^t \in \mathbb{R}^D$ and output sequences $y^t \in \mathbb{R}^N$: $\hat{h}^t = \mathbf{A}h^t + \mathbf{B}x^t$, $y^t = \mathbf{C}h^t$, where $\mathbf{A} \in \mathbb{R}^{N \times N}$ and $\mathbf{B}, \mathbf{C} \in \mathbb{R}^{N \times D}$ are learnable matrices. Due to the complexity of solving the above differential equation in deep learning settings, discrete state space models [21] discretize this system using a time-scale parameter $\mathbf{\Delta}$: $h^t = \bar{\mathbf{A}}h^{t-1} + \bar{\mathbf{B}}x^t$, $y^t = \mathbf{C}h^t$, where $\bar{\mathbf{A}} = \exp \mathbf{\Delta A}$, and $\bar{\mathbf{B}} = (\mathbf{\Delta A})^{-1}(\exp(\mathbf{\Delta A} - \mathbf{I})) \cdot \mathbf{\Delta B}$. Note that $\mathbf{\Delta}$ in discrete SSMs operates similarly to that in Eqn. 1. Discrete-time SSMs are also shown to be equivalent to the following convolution: $y = x * \bar{\mathbf{K}}$, where $\bar{\mathbf{K}} = (\mathbf{C}\bar{\mathbf{B}}, \mathbf{C}\bar{\mathbf{A}}\bar{\mathbf{B}}, \ldots, \mathbf{C}\bar{\mathbf{A}}^{L-1}\bar{\mathbf{B}})$. Therefore, the continuous form $(\mathbf{\Delta}, \mathbf{A}, \mathbf{B}, \mathbf{C})$ transitions to the discrete form $(\bar{\mathbf{A}}, \bar{\mathbf{B}}, \mathbf{C})$, allowing efficient computation using a linear recursive approach [22]. Furthermore, structured state space sequence models (S4) structures the state matrix $\mathbf{A}$ based on HIPPO matrices, significantly improving efficiency and performance.

Recently, a selective structured state space architecture named Mamba [20] was introduced. It leverages recurrent scans and a selection mechanism to control which part of the sequence flows into the hidden states. Mamba's selection mechanism can be interpreted as using data-dependent state transition mechanisms, meaning $\mathbf{\Delta}$, $\mathbf{B}$ and $\mathbf{C}$ are functions of the input $x^t$. In this work, we utilize Mamba's selection mechanism to handle irregularly sampled trajectories by learning $\mathbf{\Delta}$ from node feature set $V^t$, enhancing the method's ability to model system dynamics. By stacking Residual Blocks containing selective SSM modules, we aggregate embedded dynamics from the outputs of different blocks, allowing us to reconstruct structures with dynamics from various temporal dependencies and address incomplete node observation.

### 3.3 Generative Flow Networks

GFNs [7, 8] are generative models operating over structured sample space $\mathcal{X}$, characterized by a directed acyclic graph $\mathcal{G}$ with state space $\mathcal{S}$, where $\mathcal{X} \subseteq \mathcal{S}$. It is crucial to distinguish this from the interaction graph $G$ of the dynamical system, and the input sequence $x^t$ in SSMs. A sample $x \in \mathcal{X}$ is constructed by traversing $\mathcal{G}$ from an initial state $s_0$ to a terminal state $s_f$, the latter being a special state indicating the end of the sequence. Terminal states in $\mathcal{X}$ are those connected by a directed edge $x \to s_f$, representing valid samples of the distribution induced by GFN. A terminal state trajectory in $\mathcal{G}$ is represented by a path $s_0 \rightsquigarrow s_f$, distinct from the observational trajectories used in structural inference. Each terminal state is associated with a reward $R(X) \geq 0$, representing its unnormalized probability. The distribution over terminal states is proportional to the reward, which is governed by a flow function $F_\Omega(s \to s') \geq 0$ and satisfies the flow-matching conditions:

$$\sum_{s \in \mathrm{Pa}_\mathcal{G}(s')} F_\Omega(s \to s') - \sum_{s'' \in \mathrm{Ch}_\mathcal{G}(s')} F_\Omega(s' \to s'') = R(s'). \tag{2}$$

$\mathrm{Pa}_\mathcal{G}(s')$ and $\mathrm{Ch}_\mathcal{G}(s')$ represent the preceding state and the subsequent state of $s'$, respectively. The forward transition probability is $P(s_{k+1}|s_k) \propto F_\Omega(s_k \to s_{k+1})$, leading to a marginal probability for terminating in $x \in \mathcal{X}$ proportional to $R(x)$. Consequently, there is also a backward transition probability $P_B(\cdot)$, but it is usually set to some fixed distribution (e.g. the uniform distribution over the parent states) to reduce the search space, making the forward transition probability the only quantity to learn [7, 8, 17]. Starting from the initial state $s_0$, if we sample a terminal state trajectory $(s_0, s_1, \ldots, s_{K-1}, x, s_f)$ following the forward transition probability, defined as:

$$P(s_{k+1}|s_k) \propto F_\Omega(s_k \to s_{k+1}), \tag{3}$$

where $s_K = x$ and $s_{K+1} = s_f$, then the likelihood of a state trajectory ending in a state $x \in \mathcal{X}$ is directly proportional to the reward $R(x)$. The flow function $F_\Omega(s \to s')$, often parameterized by a neural network, is optimized to minimize discrepancies in the flow-matching conditions. This optimization results in a transition model capable of approximately sampling from the distribution over $\mathcal{X}$ in proportion to $R$. In the subsequent sections, we elucidate the construction of state spaces using graphs and detail the approximation of dynamical systems' structures through posterior estimation with a GFN. We demonstrate how GFN enhances SICSM by utilizing the embedded dynamics from Residual Blocks to reconstruct the structure of dynamical systems effectively. Moreover, SICSM facilitates the seamless integration of prior knowledge about existing connections into its training process, thereby improving the model's accuracy and efficacy in structural inference.

## 4 Structural Inference with Conjoined State Space Models

SICSM utilizes selective SSM modules, embedding in Residual Blocks, to adaptively manage input-dependent time intervals $\mathbf{\Delta}$, thus effectively handling irregularly sampled trajectories. This model architecture further aggregates dynamic embedding from multiple Residual Blocks, to obtain dynamics from various temporal dependencies, enhancing our ability to process partial observations. These dynamics are subsequently input into a GFN to approximate and sample the graph structures representing the system's structure. Figure 1 provides a schematic overview of the SICSM pipeline.

### 4.1 Aggregation of Learned Dynamics

As illustrated in the upper row of Figure 1, the primary function of the upper branch of the SICSM is time-series forecasting. These models dynamically adapt the state space parameters $(\mathbf{\Delta}, \mathbf{B}, \mathbf{C})$

Figure 1: **(Upper)** System architecture. **(Lower Left)** Detail of a Residual Block. **(Lower Right)** Structure of the Generative Flow Network for approximating the joint posterior distribution.

based on the node-specific feature embeddings. Inspired by recent advancements in dimensionality reduction and noise reduction in mixed-node feature datasets [56], we introduce a feature-based embedding network designed to compress the feature dimension from $d$ to 1:

$$\mathbf{h}_i^t = f_{embed}(v_i^t), \text{ for } t \in \{0, 1, 2, \ldots, T-2\}, \tag{4}$$

where $f_{embed}$ is a multi-layer perceptron. $\mathbf{h}_i^t$ are sequentially organized per node to form $\mathbf{H}_i = [\mathbf{h}_i^0, \mathbf{h}_i^1, ..., \mathbf{h}_i^{T-2}]$. The node dynamics are modeled by a series of Residual Blocks configured in an encoder-decoder structure. Each Residual Block incorporates a selective SSM module that dynamically learns the SSM parameters $(\mathbf{\Delta}, \mathbf{B}, \mathbf{C})$ based on the embeddings $\mathbf{H}_i$ for each node:

$$(\mathbf{\Delta}_i, \mathbf{B}_i, \mathbf{C}_i) = f_{SSMproj}(\mathbf{H_i}), \tag{5}$$

where $f_{SSMproj}$ is a linear projection layer specific to each selective SSM, as described in [20]. This allows each node's input-dependent step-size $\mathbf{\Delta}$ to adjust dynamically, enhancing the model's ability to handle irregularly sampled trajectories and reflect flexible time intervals $\mathbf{\Delta}$ as specified in Eqn. 1. Furthermore, the matrices $\mathbf{B}$ and $\mathbf{C}$ are tailored for each node, updating node features over time and accommodating the unique dynamics of each node. More details on the selection SSM in this work can be found in Appendix A.

To enhance the architectural sophistication of our model, we arrange $L$ Residual Blocks in a sequential configuration (to build a residual model), with the output of each block feeding directly into the next. This design significantly improves the model's ability to discern and interpret diverse features and aspects of the trajectory, facilitating the capture of a broad spectrum of temporal dependencies. The input to the first Residual Block is obtained by processing the concatenated node embeddings $\mathbf{H}_{all} = [\mathbf{H}_i, \text{ for all nodes}]$:

$$U_{RB\_1} = f_{RB\_1}(\mathbf{H}_{all}), \tag{6}$$

where $f_{RB\_1}$ denotes the first Residual Block function. The $l$-th Residual Block has: $U_{RB\_l} = f_{RB\_l}(U_{RB\_{l-1}})$. This setup is particularly vital in systems with partial observability, where direct connections may not be visible. In such scenarios, observable nodes may appear isolated but are frequently connected through hidden intermediaries, converting straightforward interactions into intricate multi-hop relationships. SICSM accommodates this complexity by integrating dynamics across a spectrum of temporal dependencies: shorter dependencies help reconstruct direct interactions, while longer dependencies are crucial for mapping multi-hop relationships. Unlike traditional methods that operate under fixed time intervals and predetermined direct interactions [31, 2, 57, 12, 39, 51, 54], which struggle with variable conditions, our SICSM's flexibility in adapting to different hop distances is essential. This adaptability enables it to accurately delineate potential indirect interactions and therefore to deal with incomplete observation.

Further enhancing our model, we implement an encoder-decoder structure composed of an additional $L'$ Residual Blocks. This configuration not only maintains the model's symmetry but also boosts its accuracy. The outputs from these blocks undergo transformation via a projection network, which restores the features to their original $d$-dimensional state, preparing estimated node features for the subsequent time step, $\hat{v}_i^{t+1}$. Based on the system is fully observed or not, we have:

$$U_{All} = \begin{cases} U_{RB\_L} & \text{if we observe all nodes,} \\ \sum_{l=1}^{L} U_{RB\_l} & \text{if we observe partial nodes,} \end{cases} \qquad (7)$$

$U_{RB\_l}$ represents outputs from $l$-th block. The aggregation of outputs from multiple blocks is crucial, especially in scenarios with partial observations typically caused by non-visible intermediate nodes. This aggregation ensures that dynamics of various temporal dependencies are comprehensively captured, enabling SICSM to deal with the partial observation.

## 4.2 Approximation of Posterior with a Generative Flow Network

Upon acquiring the aggregated dynamics $U_{All}$ from either the output of the final Residual Block or the summation of outputs from all Residual Blocks, we feed these dynamics to the following GFN. We utilize a GFN to model the posterior distribution $P(\mathbf{Adj} \mid U_{All})$, where $\mathbf{Adj}$ delineates the structure of the dynamical system under study. While any GFN capable of modeling the structure of graphs could be employed, we specifically choose the Joint Structure-Parameters GFN (JSP-GFN) [18] for its ability to capture the diverse dynamics of each node influenced by their interactions. This model effectively addresses the joint posterior distribution $P(\mathbf{Adj}, \lambda \mid U_{All})$, where $\lambda = \{\lambda_1, \ldots, \lambda_n\}$ represents the parameters for conditional probability distributions associated with each node $i$, enhancing the accuracy and depth of structural inference by accommodating the unique characteristics and relationships of each node.

As depicted in the lower right of Fig. 1, the state construction process begins with an initial state containing a graph with an empty adjacency matrix $G_0 = (U_{All}, \mathbf{Adj}_0)$, which progressively evolves by systematically adding edges to $\mathbf{Adj}$ based on the forward transition probability $P_\Omega(G'|G)$, where $G'$ is the resultant graph state. This iterative addition continues until a 'stop' action is chosen, signifying the completion of the graph construction phase. Once the graph $G$ is established, we proceed to generate the parameter set $\lambda$, conditioned on $G$, utilizing the forward transition probabilities $P_\Omega(\lambda|G)$. Each terminal state $s = \langle G, \lambda \rangle$ thus encapsulates a potential configuration of the system, with the construction process forming a tree structure rooted at $G_0$.

To approximate the joint posterior distribution $P(\mathbf{Adj}, \lambda|U_{All})$ which is proportional to $P(\mathbf{Adj}, \lambda, U_{All})$ [18], we define a reward function for each terminal state:

$$R(\langle G, \lambda \rangle) = P(U_{All}|\lambda, \mathbf{Adj})P(\lambda|\mathbf{Adj})P(\mathbf{Adj}), \qquad (8)$$

where $P(U_{All}|\lambda, \mathbf{Adj})$ represents the likelihood model implemented via a neural network that operates on each node, $P(\lambda|\mathbf{Adj})$ denotes the prior over the parameter set, and $P(\mathbf{Adj})$ constitutes the general prior over structures. This reward function integrates the likelihood of the observational data with the parameter and graph priors, directing the learning towards accurate structural inference. To derive the adjacency matrix for the system, we approximate the marginal posterior $P(\mathbf{Adj}|U_{All})$ by collecting samples $\{\mathbf{Adj}_1, \mathbf{Adj}_2, \ldots, \mathbf{Adj}_B\}$ from the posterior distribution and estimate the marginal probability of an edge from node $i$ to node $j$ as:

$$P_\Omega(i \to j|U_{All}) \approx \frac{1}{B} \sum_{b=1}^{B} \mathbf{1}(i \to j \in \mathbf{Adj}_b), \qquad (9)$$

where $\mathbf{1}(\cdot)$ is the indicator function. The collected value $P_\Omega$ is the approximation of structure of the dynamical system. This methodology ensures robust inference of the structure of the dynamical system. For details on the specific GFN in SICSM, please refer to Appendix B.

This approach of modeling dynamical systems using a conjoined state space model framework integrates observational trajectory modeling via selective SSMs with posterior distribution modeling via a GFN, providing a comprehensive method for structural inference in complex systems.

## 4.3 Reward Function

The reward function in SICSM, as defined in Eqn. 8, comprises three key components: the likelihood model $P(U_{All}|\lambda, G)$, the prior over parameters $P(\lambda|G)$, and the prior over graphs $P(G)$. Consistent with standard practices in GFNs [7, 8], we utilize a logarithmic transformation of the reward function:

$$\log R(\langle G, \lambda \rangle) = \log P(U_{All}|\lambda, \mathbf{Adj}) + \log P(\lambda|\mathbf{Adj}) + \log P(\mathbf{Adj}), \qquad (10)$$

with component being implemented distinctly:

**Likelihood Model.** The first term of the reward function is a log-likelihood model that estimates prediction errors for future node features, considering the current graph structure $\tilde{Adj}$ in each state. This approach aligns naturally with the evolution of dynamical systems:

$$\log P(U_{All}|\lambda, \mathbf{Adj}) = \sum_{t=0}^{T-2} \sum_{n=1}^{N} \log P(v_i^{t+1}|\lambda, \tilde{Adj}, U_i^t), \tag{11}$$

where $U_i^t$ is the learned embeddings of node $i$ at time $t$ and is obtained from $U_{All}$. $\log P(v_i^{t+1}|\lambda, \tilde{Adj}, U_i^t)$ is modeled with a neural network to enable accurate predictions of future node features. This setup integrates both node features and graph structure into the computation, ensuring their collective influence on the reward function.

**Parameter Prior.** The second term of the reward function represents the prior over the parameters $\lambda$. We utilize a unit Normal distribution for each parameter $\lambda_{ij}$, correlating with the sender and receiver nodes indexed by $i$ and $j$: $P(\lambda_{ij}|\mathbf{Adj}) = \mathcal{N}(0,1)$. This choice of prior contributes to a balanced modeling of node interactions within the graph.

**Graph Prior.** The component of the graph prior comprises a uniform prior alongside regularization terms designed to enhance the graph's structural smoothness. In reference to the current graph structure $\tilde{Adj}$, it is formulated as:

$$P(\mathbf{Adj}) = P_U(\mathbf{Adj}) + \exp\left(D(\tilde{Adj}, U_{All}) + \mathcal{L}_d(\tilde{Adj}) + \mathcal{L}_s(\tilde{Adj})\right), \tag{12}$$

where $P_U(\mathbf{Adj})$ is the uniform prior. The regularization terms include Dirichlet energy $D(\tilde{Adj}, U_{All})$ to measure smoothness between adjacent node features, a connectivity term $\mathcal{L}_d(\tilde{Adj})$ to penalize unconnected structures, and a sparsity term $\mathcal{L}_s(\tilde{Adj})$ to regulate graph density:

$$D(\tilde{Adj}, U_{All}) = -\frac{1}{n^2} \sum_{i,j} \tilde{Adj}_{ij} \|U_i - U_j\|^2, \tag{13}$$

$$\mathcal{L}_d(\tilde{Adj}) = \frac{1}{n} \mathbf{1}^{\mathsf{T}} \log(\tilde{Adj}\mathbf{1}), \text{ and } \mathcal{L}_s(\tilde{Adj}) = -\frac{1}{n^2} \|\tilde{Adj}\|_F^2. \tag{14}$$

These terms, adapted for the reward function in SICSM, emphasize the influence of the graph's properties on the state space, enriching the model's structural inference capability.

## 4.4 Learning Objectives

The learning objectives of SICSM are bifurcated into two main components: (1) time-series forecasting using Residual Blocks, and (2) modeling dynamics with the GFNs, focusing on the accuracy of transition probabilities. For the Residual Blocks, the primary learning objective is the minimization of the Mean Squared Error between predicted and actual node features across all time steps and nodes:

$$\mathcal{L}_{RB} = \frac{1}{T \times n} \sum_{t=0}^{T-2} \sum_{i=1}^{n} \|v_i^{t+1} - \hat{v}_i^{t+1}\|^2, \tag{15}$$

where $\hat{v}_i^{t+1}$ and $v_i^{t+1}$ represent the predicted and actual features of node $i$ at time $t+1$, respectively. This objective ensures that the Residual Blocks effectively capture and forecast the dynamics. Similar to [17, 18], the objective for the GFN involves optimizing the squared error of the logarithmic ratio between forward and backward transition probabilities to ensure accurate modeling of the graph structure:

$$\mathcal{L}_{GFN} = \mathbb{E}_\pi \left[ \left( \log \frac{R(\langle G', \wedge\lambda'\rangle) P_B(G|G') P_\Omega(\wedge\lambda|G)}{R(\langle G, \wedge\lambda\rangle) P_\Omega(G'|G) P_\Omega(\wedge\lambda'|G')} \right)^2 \right], \tag{16}$$

where $P_B(\cdot)$ indicates the backward transition probability, and $\lambda'$ denotes the parameters generated conditional on graph $G'$. The sampling distribution $\pi$ covers pairs $\langle G, \lambda\rangle$ and $\langle G', \lambda'\rangle$, with the 'stop-gradient' operation ($\wedge$) critical for halting backpropagation through the parameters $\lambda$ and $\lambda'$, thus preventing potential feedback loops during training. Please refer to Appendix B for the

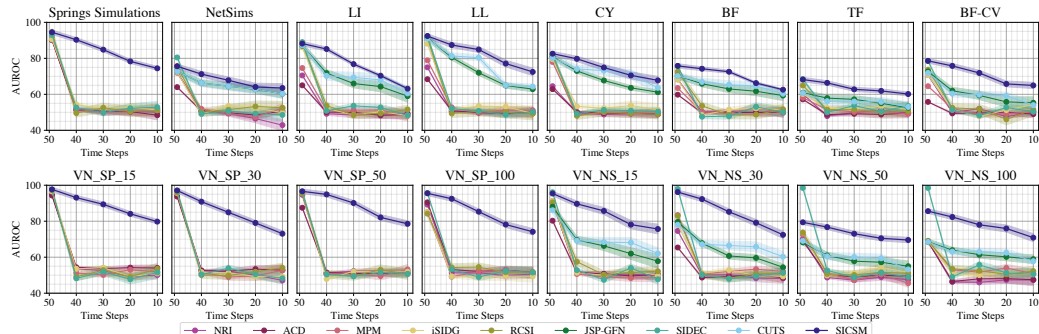

Figure 2: AUROC values (expressed in percentage) for various methods as a function of the number of irregularly sampled time steps. Results are averaged across ten trials, with time steps varying from 49 to 10. All subplots share a common x-axis and y-axis for uniform comparison.

parameterization of these terms. The final learning objective of SICSM combines these terms:

$$\mathcal{L} = \mathcal{L}_{RB} + \mathcal{L}_{GFN}. \tag{17}$$

This combined objective facilitates concurrent optimization of both time-series prediction accuracy and the fidelity of the inferred graph structure.

## 4.5 Integration of Prior Knowledge

In SICSM, the integration of prior knowledge concerning existing network connections is executed more seamlessly and effectively compared to traditional VAE-based methods [31, 39, 51, 54, 53]. To incorporate this prior knowledge, we initialize the graph $G_0$ in the GFN's initial state with edges that represent the known connections based on prior knowledge: $G_{0,k} = (\mathcal{V}, \mathbf{Adj}_0 \cup E_k)$, where $E_k$ contains the set of known edges based on prior knowledge. This setup ensures that the learning and sampling processes are continually influenced by this integrated knowledge, enhancing the model's accuracy and effectiveness in predicting and understanding the underlying dynamics of the system.

# 5   Experimental Results

This study systematically evaluates the performance of SICSM across an extensive array of datasets, which encompass both one-dimensional and multi-dimensional trajectories. The investigation specifically concentrates on challenging scenarios of irregularly sampled trajectories and partial observations. More results and methodological specifics are further elaborated in Appendix E.

## 5.1   General Settings

**Datasets.** Our study first evaluates the SICSM model on two established structural inference datasets: the Spring Simulations dataset [31], which simulates dynamic interactions of balls connected by springs within a symmetric setting, and the NetSim dataset [47], which consists of simulated blood-oxygen-level-dependent imaging data from various brain regions in an asymmetric network. Both datasets include 10 nodes, with Spring Simulations offering four-dimensional features and NetSim one-dimensional features at each timestep, initially sampled at 49 regular intervals.

Additionally, we examined six directed synthetic biological networks (Linear, Linear Long, Cycle, Bifurcating, Trifurcating, and Bifurcating Converging) as outlined in [44], with abbreviations LI, LL, CY, BF, TF and BF-CV, respectively. These networks simulate developmental trajectories in differentiating cells using BoolODE [44], capturing one-dimensional mRNA expression levels over 49 timesteps with irregular intervals tailored to our experimental setups.

We also incorporated data from the StructInfer Benchmark [3], focusing on 'Vascular Networks' (VN) with node counts ranging from 15 to 100. These datasets, named under the categories Springs (SP) and NetSims (NS), were selected for their complex and varying underlying graph structures, providing a robust platform to validate the efficacy of the SICSM model.

**Baselines and metrics.** To evaluate the performance of SICSM, we compared it against a suite of state-of-the-art models: NRI [31], MPM [12], ACD [39], iSIDG [51], RCSI [54], JSP-GFN [18], CUTS [13], and SIDEC [53]. The comparative effectiveness of these methods was quantitatively assessed using the area under the receiver operating characteristic (AUROC) curve, focusing on the accuracy of the inferred adjacency matrix relative to the ground truth.

**Experimental settings.** All experiments were conducted on a single NVIDIA Ampere 40GB HBM graphics card, paired with 2 AMD Rome CPUs (32 cores@2.35 GHz). Detailed configurations and additional results are elaborated in Appendix D and Appendix E.

## 5.2   Experimental Results with Irregularly Sampled Trajectories

This section examines the performance of the evaluated methods to irregular sampling of input trajectories. Detailed in Section 5.1, our datasets undergo randomized reduction in time steps to $[40, 30, 20, 10]$ from an original count of $49$. The baselines, alongside SICSM, are then trained and evaluated on these irregularly sampled trajectories, with the average AUROC results of 10 runs depicted in Figure 2. Besides, we report the AUPRC results, SHD values and F1-scores in Figues 7-9 in Appendix E.1. It should be noted that JSP-GFN, being limited to one-dimensional feature analysis, is not applicable to multi-dimensional datasets such as Springs Simulations and VN_SP.

SICSM exhibits exceptional consistency in its performance despite the decrease in time steps, which is a critical indicator of robustness within structural inference models. In datasets like Spring Simulations and NetSim, while the baseline models show significant declines in performance from 49 to 10 time steps, SICSM maintains AUROC scores above 85%. This underscores its potent capability to effectively leverage essential structural information, even when data availability is constrained. Moreover, when faced with irregular sampling, traditional VAE-based methods such as NRI, ACD, MPM, iSIDG, and RCSI struggle significantly, often performing no better than random guessing. This decline is primarily attributed to their dependency on fixed time intervals between observations—an assumption not held in our experimental conditions. In contrast, in challenging synthetic biological networks like BF and TF, SICSM consistently surpasses baseline models by margins of 5-10% across all sampling levels, affirming its sophisticated understanding of complex, directional interactions which are crucial in genomics and systems biology.

SICSM's robustness to irregular sampling intervals is particularly notable in datasets such as VN_SP_30 and VN_SP_50. Unlike conventional models that falter under variable data availability, SICSM's architecture, equipped with adaptive time-interval handling, adeptly navigates these challenges, preserving its predictive accuracy. These findings validate the effectiveness of SICSM in managing complex, temporally variant structural inference challenges across a spectrum of demanding datasets. The model not only demonstrates resilience to data scarcity and irregular sampling, but also excels in capturing intricate systemic interactions.

## 5.3   Experimental Results with Incomplete Observation of Systems

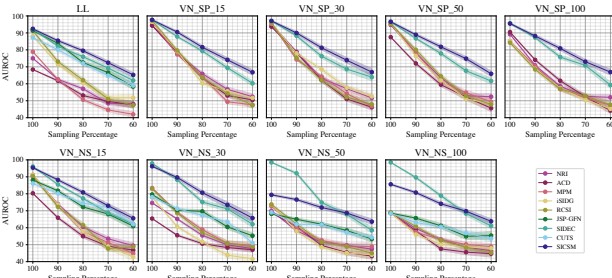

Figure 3: AUROC values (expressed in percentage) for various methods as a function of the proportion of observed nodes, averaged over 20 trials. Node sampling proportions are set at $[100\%, 90\%, 80\%, 70\%, 60\%]$.

This section explores the resilience of evaluated methods to scenarios where only a subset of the system's nodes is observable. As outlined in Section 5.1, the datasets undergo a reduction in node count by sampling from all nodes, scaled to proportions of $[100\%, 90\%, 80\%, 70\%, 60\%]$, with rounding up to ensure integer counts. This experimental setup was applied particularly to datasets with more than 10 nodes—specifically the LL and all VN datasets—to facilitate a comprehensive investigation. The performance of the baselines, alongside our SICSM model, is quantified using the average AUROC from 20 runs, as depicted in Figure 3.

SICSM demonstrates remarkable stability in AUROC scores across different levels of node sampling, particularly excelling in environments with partial observations. In the VN_SP_30 dataset, it maintains an AUROC above 80%, even with only 60% of nodes observed, significantly outperforming methods like NRI and ACD, whose performance dips below 75%. This highlights SICSM's ability to effectively utilize essential structural relationships under partial observations. In complex network structures like the VN_NS series, SICSM's strong performance underscores its proficiency in inferring critical interactions, despite considerable reductions in observable nodes. This robustness showcases the model's ability for managing data sparsity and leveraging available information effectively.

SIDEC also shows competitive performance, underscoring the benefits of dynamics-encoding models in handling incomplete observations. However, without an adaptive transition dynamic function, SIDEC generally underperforms compared to SICSM, especially when fewer nodes are observed. In comparisons, SICSM consistently outshines JSP-GFN at lower node sampling percentages, illustrating its superior capability in managing partial observations and effectively using structural information even with limited data visibility. These results confirm SICSM's robustness in structural inference, particularly in scenarios with incomplete observations. Its resilience to node sparsity and ability to discern complex interactions make it a valuable tool for applications that require reliable, accurate structural predictions in data-constrained environments.

### 5.4 Why Do We Need All Residual Outputs?

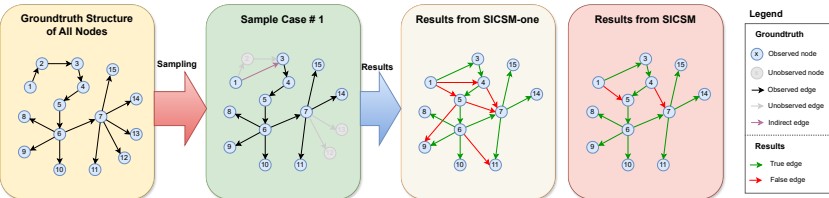

Figure 4: **(1st Column)** Ground truth structure with all nodes. **(2nd Column)** One example of 12-node sampling. **(3rd, 4th Columns)** Structural inference results from SICSM-one and SICSM.

As discussed in Section 3.2, systems with partial observations can benefit from aggregating dynamics from multiple Residual Blocks for GFN input. We explored this by analyzing the use of only the dynamics from the final Residual Block of encoder in a case study using the VN_SP_15 dataset, where 12 nodes (80% of the total) are sampled. We evaluated two configurations: the comprehensive SICSM, integrating outputs from all Blocks in encoder, and SICSM-one, which relies solely on the final Block's output. Figure 4 shows that SICSM-one often inaccurately classified two-hop interactions as three-hop connections, leading to increased false positives due to its dependence on the output from the larger, final Residual Block, which tends to blur hop distinctions. In contrast, SICSM's approach of using outputs from multiple layers provided a rich dynamics representation that effectively managed both shorter and longer connections, reducing wrong results. Despite these strengths, some inaccuracies point to the potential need for an adaptive weighting mechanism that adjusts the influence of dynamics based on the graph's size and longest paths, potentially improving accuracy across different scenarios. These results, along with an additional case in Appendix E.4, highlight the effectiveness of SICSM's multi-layer dynamic integration in handling partial observations and its capability for precise structural inference in complex settings.

## 6 Conclusion

This paper presents SICSM, a novel structural inference approach that merges Selective State Space Models with Generative Flow Networks. By embedding dynamics with Residual Blocks, our method learns input-dependent transition parameters, effectively handling irregularly sampled trajectories. Aggregating outputs from multiple blocks enriches the dynamics captured, addressing the significant challenge of incomplete node observations. The downstream Generative Flow Network, leveraging these dynamics, achieves precise structural inference and seamlessly incorporates prior knowledge. Empirical evidence demonstrates SICSM's effectiveness, particularly in settings with irregular sampling and partial observations. Future research will explore specific adaptations of conjoined state space models for dynamic systems with mutable structural elements, such as evolving connections and emerging nodes. Additionally, we aim to explore the development of a comprehensive model capable of pioneering new paths in general scientific discovery.

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

# Appendix of Structural Inference of Dynamical Systems with Conjoined State Space Models

## A More Details on Selective SSM in SICSM

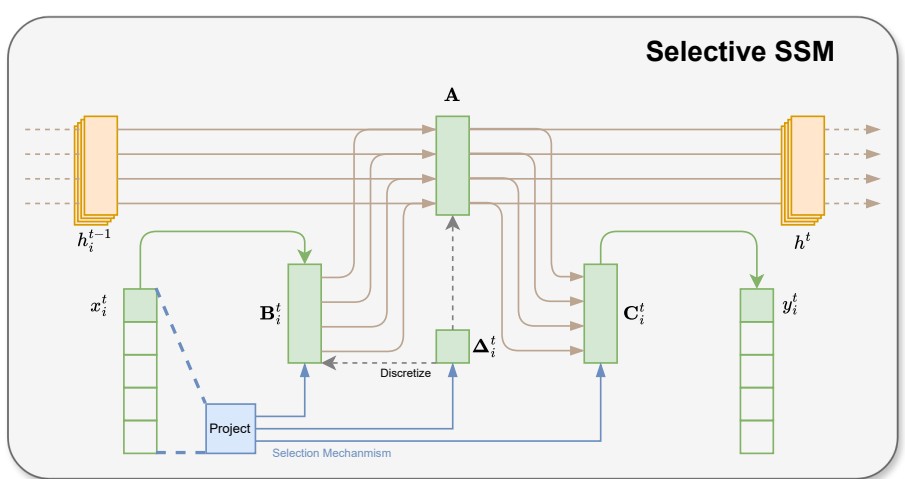

Figure 5: The overview of Selective SSM in Residual Blocks.

In this section, we delve deeper into the capabilities of the Selective SSM, which is central to our SICSM framework, enabling it to learn input-dependent time intervals, denoted as $\mathbf{\Delta}$, with enhanced adaptability. As illustrated in Figure 5, the selective SSM processes time-series data for each node, represented by $x_i$. For each node $i$ and at each time step $t$, the projection layer of the selective SSM dynamically learns the transition parameters $\mathbf{B}_i^t$, $\mathbf{C}_i^t$, and $\mathbf{\Delta}_i^t$. This capability not only provides the flexibility needed to adapt to varying time intervals but also significantly enriches the model's ability to capture the nuanced dynamics of each node over time.

Contrasting with previous models that often rely on static or less adaptable transition parameters [31, 58, 39, 51], our selective SSM design allows SICSM to adjust its learning mechanism based on the input data's temporal characteristics. This flexibility is crucial for effectively modeling the transitions in dynamics, particularly when dealing with irregularly sampled trajectories. By enabling the selective SSM to adapt its parameters dynamically, SICSM can more accurately reflect the evolving dynamics inherent in complex systems, thus providing a robust framework for predicting changes and interactions within these systems under varying observational conditions.

## B More Details on GFN in SICSM

The discussion in this section is an addition to the description of GFN used in SICSM (in Section 4.2), with an illustration of the GFN shown in Figure 6.

### B.1 More Details on Flow-matching Conditions

We first provide more introduction on general GFNs. GFNs, originally conceptualized through the flow-matching conditions as proposed by Bengio et al. [7], have seen the development of alternative

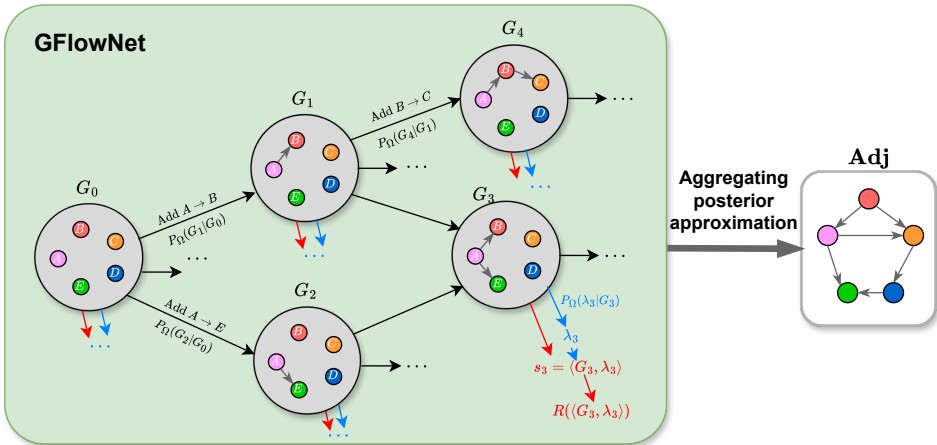

Figure 6: Structure of GFN in SICSM. Each state $s$ consists of a graph structure $G$ of the underlying interaction graph of the dynamical system, and a generated parameter $\lambda$. The initial state $s_0$ is the completely disconnected graph. Each state $s$ is complete and connected to a terminal state $s_f$ and associated to a reward $R(\langle G, \lambda \rangle)$. Transitioning from one state to another corresponds to adding a directed edge to the graph.

conditions that ensure equivalent guarantees. These conditions ensure that a GFN satisfying them would sample complete states in proportion to their associated rewards.

One such alternative is the detailed balance conditions (DB), derived from Markov chain theory, as discussed by Bengio et al. [8]. These conditions are defined for any transition $s \rightarrow s'$ within the GFN as:

$$F(s)P_F(s'|s) = F(s')P_B(s|s'), \tag{18}$$

where $F(s)$ represents a flow function, which can be parameterized by a neural network. Bengio et al. [8] demonstrated that adherence to these detailed balance conditions across all transitions $s \rightarrow s'$ in the GFN ensures that the resulting distribution is proportional to the reward $R(s)$. In situations where all states in the GFN are complete, Deleu et al. [17] adapted these conditions to eliminate the need for a separate flow function.

An alternative set of conditions, known as trajectory balance conditions (TB), was introduced by Malkin et al. [42]. These conditions apply at the level of complete trajectories rather than individual transitions. For a complete trajectory $\tau = (s_0, s_1, ..., s_T, s_f)$, the trajectory balance condition is formulated as:

$$Z \prod_{t=1}^{T} P_F(s_{t+1}|s_t) = R(s_T) \prod_{t=1}^{T-1} P_B(s_t|s_{t+1}), \tag{19}$$

with the convention $s_{T+1} = s_f$, and where $Z$ is the partition function of the distribution (i.e., $Z = \sum_{x \in \mathcal{X}} R(x)$); in practice, $Z$ is a learnable parameter of the model that is being learned alongside the forward and backward transition probabilities. Compliance with the trajectory balance conditions across all complete trajectories ensures that the induced distribution by the GFN is proportional to the reward $R(s)$.

## B.2 Learning Objective of GFN

The GFN of the SICSM learning framework adopts the Subtrajectory Balance conditions (SubTB) [42], a critical concept for ensuring the balance of flow in GFNs, which is a more general and relaxed condition than DB. Instead of enforcing balance at each state transition, SubTB focuses on balancing the probability mass over entire subtrajectories of the generative process. A detailed overview of SubTB is provided in Appendix B.3. The GFN in SICSM is characterized by a dual structure, comprising both the interaction graph $G$ and the node-specific parameters $\lambda$. To accommodate

this unique structure, we adopt a modified form of SubTB as proposed in [18]:

$$R(\langle G', \lambda' \rangle) P_B(G|G') P_\Omega(\lambda|G) = R(\langle G, \lambda \rangle) P_\Omega(G'|G) P_\Omega(\lambda'|G'), \tag{20}$$

where $P_B(\cdot)$ represents the backward transition probability, and $\lambda'$ denotes the parameters generated conditional on graph $G'$. This reformulation ensures that both the graph $G$ (including the aggregated node dynamics embeddings $U_{All}$ as well as the structure $\mathbf{Adj}$) and the parameters $\lambda$ are integral to the reward function, thereby reinforcing their importance in structural inference tasks. We show in Appendix E.2 with experimental results that this set up greatly encourages the successful integration of prior knowledge on existing edges. Additional details on this reformulation are available in Appendix B.4.

For SICSM, we define the learning objective as the squared error of the log ratio between forward and backward transitions, in line with the approach in [18]. Specifically, the learning objective is given by:

$$\mathcal{L}_{GFN} = \mathbb{E}_\pi \left[ \left( \log \frac{R(\langle G', \wedge\lambda' \rangle) P_B(G|G') P_\Omega(\wedge\lambda|G)}{R(\langle G, \wedge\lambda \rangle) P_\Omega(G'|G) P_\Omega(\wedge\lambda'|G')} \right)^2 \right], \tag{21}$$

where $\pi$ is a sampling distribution over pairs $\langle G, \lambda \rangle$ and $\langle G', \lambda' \rangle$, and $\wedge$ denotes the 'stop-gradient' operation. This operation is crucial to prevent backpropagation through $\lambda$ and $\lambda'$, thereby avoiding potential infinite loops.

### B.3  More details about SubTB

The concept of subtrajectory balance conditions, introduced by Malkin et al. [42], serves as a generalization of both detailed balance and trajectory balance conditions, extending their application to partial trajectories of varying lengths. These conditions are defined for a partial state trajectory $\tau = (s_m, s_{m+1}, ..., s_n)$ as follows:

$$F(s_m) \prod_{t=m}^{n-1} P_F(s_{t+1}|s_t) = F(s_n) \prod_{t=m}^{n-1} P_B(s_t|s_{t+1}), \tag{22}$$

where $F(s)$ denotes a flow function. This framework effectively encapsulates both conditions outlined in Appendix B.1, by accommodating for partial state trajectories of single-step transitions (as in Eqn. 18), and for complete trajectories (as in Eqn.19), with $F(s_0) = Z$ as per Bengio et al. [8]). Furthermore, Madan et al. [41] proposed a novel objective, SubTB($\lambda$), which synergizes subtrajectory balance conditions for partial trajectories of differing lengths, drawing inspiration from the TD($\lambda$) approach in reinforcement learning.

These subtrajectory balance conditions are also adaptable to undirected paths, allowing for "back and forth" movements between states [42]. For an undirected path between $s_m$ and $s_n$, this (generalized) subtrajectory balance condition can be written as

$$F(s_m) \prod_{t=k}^{m-1} P_B(s_t|s_{t+1}) \prod_{t=k}^{n-1} P_F(s_{t+1}|s_t) = F(s_n) \prod_{t=k}^{n-1} P_B(s_t|s_{t+1}) \prod_{t=k}^{m-1} P_F(s_{t+1}|s_t), \tag{23}$$

where $s_k$ is a common ancestor of both $s_m$ and $s_n$. These conditions, whether generalized or specific, offer greater flexibility in their application. However, to guarantee that a GFN induces a distribution proportional to $R(s)$, it is essential that these conditions are satisfied for all partial trajectories of any length. In this paper, we specifically focus on scenarios where these conditions are met for partial state trajectories of fixed length. Although this approach may deviate from the general guarantees, we follow the implementation discussed by Deleu et al. [18] and expound in Appendix B.4 how our GFN still induces a distribution $\propto R(s)$ in our context.

### B.4  More details about Reformulation of SubTB

In this section, we strictly follow the implementation and proofs discussed in [18].

**Subtrajectory balance conditions for undirected paths of length 3.**  Consider an undirected path of length 3 denoted as $\langle G, \lambda \rangle \leftarrow \langle G, \cdot \rangle \rightarrow \langle G', \cdot \rangle \rightarrow \langle G', \lambda' \rangle$, where $G'$ is derived from the directed acyclic graph (DAG) $\mathcal{G}$ by the addition of a new edge.. Given that the state $\langle G, \cdot \rangle$ is a

common ancestor to both complete states $\langle G, \lambda \rangle$ and $\langle G', \lambda' \rangle$, we can apply the subtrajectory balance conditions as expressed in Eqn. 23. The conditions are reformulated as follows:

$$F(G, \lambda) P_B(G|\lambda) P_F(G'|G) P_F(\lambda'|G') = F(G', \lambda') P_B(G'|\lambda') P_B(G|G') P_F(\lambda|G), \quad (24)$$

where $P_B(G|\lambda)$ denotes $P_B(\langle G, \cdot \rangle | \langle G, \lambda \rangle)$, a notation simplification for clarity. As $\langle G, \lambda \rangle \in \mathcal{X}$ has a single parent state $\langle G, \cdot \rangle$, it follows that $P_B(G|\lambda) = 1$, and a similar rationale applies to $\langle G', \lambda' \rangle$. Building upon the insights from Deleu et al. [17], the flow function $F(G, \lambda)$ of a complete state $\langle G, \lambda \rangle$ can be expressed as a function of its associated reward:

$$F(G, \lambda) = \frac{R(\langle G, \lambda \rangle)}{P_F(s_f | \langle G, \lambda \rangle)}. \quad (25)$$

In our GFN implementation, $s_f$ is the sole child of the terminal state $\langle G, \lambda \rangle \in \mathcal{X}$, indicating an (infinitely wide) tree structure rooted at $\langle G, \cdot \rangle$. Consequently, $P_F(s_f | \langle G, \lambda \rangle) = 1$, leading to the simplification $F(G, \lambda) = R(\langle G, \lambda \rangle)$. With these simplifications, Eqn. 24 becomes

$$R(\langle G, \lambda \rangle) P_F(G'|G) P_F(\lambda'|G') = R(\langle G', \lambda' \rangle) P_B(G|G') P_F(\lambda|G), \quad (26)$$

which is the subtrajectory balance condition in Eqn. 20. This formulation effectively captures the essence of the balance conditions, providing a clear and concise representation of the underlying principles in the GFN structure for structural inference.

**Integrating undirected paths of length 2.** Similar to the previous paragraph, we consider here an undirected path of length 2 of the form $\langle G, \lambda \rangle \leftarrow \langle G, \cdot \rangle \rightarrow \langle G, \tilde{\lambda} \rangle$. Since $\langle G, \cdot \rangle$ is a common ancestor (a common parent in this case) of both terminal states $\langle G, \lambda \rangle$ and $\langle G, \tilde{\lambda} \rangle$, we can write the subtrajectory balance conditions (Eqn. 23) as:

$$F(G, \lambda) P_B(G|\lambda) P_F(\tilde{\lambda}|G) = F(G, \tilde{\lambda}) P_B(G|\tilde{\lambda}) P_F(\lambda|G). \quad (27)$$

Using the same simplifications as in the previous paragraph ($P_B(G|\lambda) = P_B(G|\tilde{\lambda}) = 1$), we get the following subtrajectory balance conditions for the undirected paths of length 2:

$$R(\langle G, \lambda \rangle) P_F(\tilde{\lambda}|G) = R(\langle G, \tilde{\lambda} \rangle) P_F(\lambda|G). \quad (28)$$

Note that these conditions are effectively redundant if the SubTB conditions over undirected paths of length 3 are satisfied for all possible pairs of terminal states $\langle G, \lambda \rangle$ and $\langle G', \lambda' \rangle$. Indeed, if we write these conditions between $\langle G, \lambda \rangle$ and $\langle G', \lambda' \rangle$ on the one hand, and between $\langle G, \tilde{\lambda} \rangle$ and $\langle G', \lambda' \rangle$ on the other hand (with a fixed $G'$ and $\lambda'$:

$$R(\langle G', \lambda' \rangle) P_B(G|G') P_F(\lambda|G) = R(\langle G, \lambda \rangle) P_F(G'|G) P_F(\lambda'|G), \quad (29)$$

$$R(\langle G', \lambda' \rangle) P_B(G|G') P_F(\tilde{\lambda}|G) = R(\langle G, \tilde{\lambda} \rangle) P_F(G'|G) P_F(\lambda'|G), \quad (30)$$

we get the same subtrajectory balance conditions over undirected paths of length 2 as in Eqn. 28:

$$\frac{R(\langle G, \lambda \rangle)}{P_F(\lambda|G)} = \frac{R(\langle G', \lambda' \rangle) P_B(G|G')}{P_F(G'|G) P_F(\lambda'|G')} = \frac{R(\langle G, \lambda' \rangle)}{P_F(\tilde{\lambda}|G)}. \quad (31)$$

However, since the SubTB conditions are only satisfied approximately in practice, it might be advantageous to also satisfy Eqn. 28. The equation above provides an alternative way to express Eqn. 28. Indeed, Eqn. 31 shows that the function

$$f_G(\lambda) \stackrel{\triangle}{=} \log R(\langle G, \lambda \rangle) - \log P_F(\lambda|G) \quad (32)$$

is constant, albeit with a constant that depends on the graph $G$. Since this function is differentiable, this is equivalent to $\nabla_\lambda f_G(\lambda) = 0$, and therefore we get the differential form of the subtrajectory balance conditions:

$$\nabla_\lambda \log P_F(\lambda|G) = \nabla_\lambda \log R(\langle G, \lambda \rangle). \quad (33)$$

As shown by [18], one way to enforce the SubTB conditions over undirected paths of length 3 is to create a learning objective that encourages these conditions to be satisfied, and optimizing it using gradient methods. The learning objective has the form $\mathcal{L}_{GFN} = \mathbb{E}_\pi[\tilde{\triangle}^2(\Omega)]$, where $\tilde{\triangle}(\Omega)$ is a non-linear residual term

$$\tilde{\triangle}(\Omega) = \log \frac{R(\langle G', \lambda' \rangle) P_B(G|G') P_\Omega(\lambda|G)}{R(\langle G, \lambda \rangle) P_\Omega(G'|G) P_\Omega(\lambda'|G')}. \quad (34)$$

Suppose that the parameters $\Omega$ of the GFN are such that the subtrajectory balance conditions in Eqn. 33 are satisfied for any $\langle G, \lambda \rangle$. Although this assumption is unlikely to be satisfied in practice, they will eventually be approximately satisfied over the course of optimization, given the discussion above about the relation between Eqn. 28 and Eqn. 26. Since $\lambda$ and $\lambda'$ depend on $\Omega$ (via the reparametrization trick since they are sampled on-policy [18]), taking the derivative of $\tilde{\triangle}^2(\Omega)$, we get:

$$\frac{d}{d\Omega}\tilde{\triangle}^2(\Omega) = \tilde{\triangle}(\Omega) \cdot \frac{d}{d\Omega}[\log R(\langle G', \lambda' \rangle) + \log P_\Omega(\lambda|G) - \log R(\langle G, \lambda \rangle) - \log P_\Omega(G'|G)$$
$$- \log P_\Omega(\lambda'|G')]. \quad (35)$$

Using the law of total derivatives, we have

$$\frac{d}{d\Omega}[\log P_\Omega(\lambda|G) - \log R(\langle G, \lambda \rangle) = \underbrace{\left[\frac{\partial}{\partial \lambda}\log P_\Omega(\lambda|G) - \frac{\partial}{\partial \lambda}\log R(\langle G, \lambda \rangle)\right]}_{=0}\frac{d\lambda}{d\Omega} \quad (36)$$

$$+ \frac{\partial}{\partial \Omega}\log P_\Omega(\lambda|G) \quad (37)$$

$$= \frac{\partial}{\partial \Omega}\log P_\Omega(\lambda|G) \quad (38)$$

and similarly for the terms in $\langle G', \lambda' \rangle$. The derivative of the objective then becomes

$$\frac{d}{d\Omega}\tilde{\triangle}^2(\Omega) = \tilde{\triangle}(\Omega) \cdot \left[\frac{\partial}{\partial \Omega}\log P_\Omega(\lambda|G) - \frac{\partial}{\partial \Omega}\log P_\Omega(\lambda'|G') - \frac{d}{d\Omega}\log P_\Omega(G'|G)\right]. \quad (39)$$

An alternative way to obtain the same derivative in Eqn. 39 as the objective in Eqn. 33 is to take $d\lambda/d\Omega = 0$ instead, meaning that we would not differentiate through $\lambda$ (and $\lambda'$). Using the stop-gradient operation $\wedge$, this shows the following objective

$$\mathcal{L}_{GFN} = \mathbb{E}_\pi\left[\left(\log \frac{R(\langle G', \wedge \lambda' \rangle)P_B(G|G')P_\Omega(\wedge \lambda|G)}{R(\langle G, \wedge \lambda \rangle)P_\Omega(G'|G)P_\Omega(\wedge \lambda'|G')}\right)^2\right], \quad (40)$$

takes the same value and has the same gradient (Eqn. 39) as the objective in Eqn. 34 when the subtrajectory balance conditions (in differential form) over undirected paths of length 2 are satisfied.

While optimizing Eqn. 40 alone leads to eventually satisfying the subtrajectory balance conditions over undirected paths of length 2, it may be advantageous to explicitly encourage this behavior, especially in cases for non-linear models. We can incorporate some penalty to the loss function, such as

$$\tilde{\mathcal{L}}_{GFN} = \mathcal{L}_{GFN} + \frac{\beta}{2}\mathbb{E}_\pi[\| \nabla_\lambda \log P_\Omega(\lambda|G) - \nabla_\lambda \log R(\langle G, \lambda \rangle)\|^2 + \| \nabla_{\lambda'} \log P_\Omega(\lambda'|G')$$
$$- \nabla_{\lambda'}\log R(\langle G', \lambda' \rangle)\|^2]. \quad (41)$$

## B.5    Forward Transition Probabilities

As delineated in Section 4.2, SICSM generates the pair $\langle G, \lambda \rangle$ through a two-phase process: (a) constructing the graph $G = (U_{All}, \mathbf{Adj})$ by sequentially adding edges in $\mathbf{Adj}$ until a 'stop' action is triggered, followed by (b) sampling the parameters $\lambda$ conditional on $G$. These actions are governed by the forward transition probabilities $P_\Omega(G'|G)$ in the first phase and $P_\Omega(\lambda|G)$ in the second phase.

To parameterize these terms, we adopt a hierarchical model strategy [18]. This model first determines whether to halt the first phase using the probability $P_\Omega(\text{stop}|G)$. Based on this decision, the process either continues by adding an edge to $\mathbf{Adj}$, forming $G' = (U_{All}, \mathbf{Adj}')$ with probability $P_\Omega(G'|G, \neg\text{stop})$, or transitions to the second phase by sampling $\lambda$ with probability $P_\Omega(\lambda|G, \text{stop})$:

$$P_\Omega(G'|G) = (1 - P_\Omega(\text{stop}|G))P_\Omega(G'|G, \neg\text{stop}), \quad (42)$$
$$P_\Omega(\lambda|G) = P_\Omega(\text{stop}|G)P_\Omega(\lambda|G, \text{stop}). \quad (43)$$

To accurately parameterize $P_\Omega(\text{stop}|G)$, $P_\Omega(G'|G, \neg\text{stop})$, and $P_\Omega(\lambda|G, \text{stop})$, we utilize a combination of graph neural networks (GNNs) [5] and self-attention mechanisms [50]. This fusion of GNNs

and self-attention blocks ensures a robust and flexible modeling of the transitions between states in SICSM. Further details regarding this parameterization approach are elaborated in Section B.6.

## B.6  Parameterization with Neural Networks

In the GFN of SICSM, various components are parameterized using neural networks. Specifically, we focus on parameterizing the following: **(a)** $P_\Omega(\text{stop}|G)$, **(b)** $P_\Omega(G'|G, \neg\text{stop})$, **(c)** $P_\Omega(\lambda|G, \text{stop})$, and **(d)** the log-likelihood term $\log P(v_i^{t+1}|\lambda, \tilde{Adj}, U_i^t)$. We combine GNN with self-attention mechanisms for parameterizing **(a)** $P_\Omega(\text{stop}|G)$ and **(b)** $P_\Omega(G'|G, \neg\text{stop})$. This process generates a graph-level attribute $\mathbf{g}$ and node-level attributes $\{\mathbf{u}_i, \mathbf{v}_i, \mathbf{w}_i\}$ for each node $i$ in $G$:

$$\mathbf{g}, \{\mathbf{u}_i, \mathbf{v}_i, \mathbf{w}_i\}_{i=1}^n = \text{SelfAttention}_\Omega\left(\text{GNN}_\Omega(G)\right). \tag{44}$$

The 'stop' action probability is computed as $P_\Omega(\text{stop}|G) = f_\Omega(\mathbf{g})$, where $f_\Omega$ is a neural network with a sigmoid output layer. The probability of transitioning from $G$ to $G'$ in the absence of a 'stop' action is defined as:

$$P_\Omega(G'|G, \neg\text{stop}) \propto \mathbf{m}_{ij} \exp(\mathbf{u}_i^\mathsf{T} \mathbf{v}_j), \tag{45}$$

where $\mathbf{m}_{ij}$ is a binary mask that excludes already explored graph structures. **(c)** For sampling parameters $\lambda_i$ for each node $i$, we define:

$$P_\Omega(\lambda_i|G, \text{stop}) = \mathcal{N}\left(\lambda_i|\mu_\Omega(\mathbf{w}_i), \sigma_\Omega^2(\mathbf{w}_i)\right), \tag{46}$$

with $\mu_\Omega$ and $\sigma_\Omega^2$ being neural networks. This formulation effectively approximates the posterior distribution $P(\lambda_i|G, U_{All})$ upon full training. To get the adjacency matrix for the dynamical system, we need to approximate the marginal posterior $P(\mathbf{Adj}|U_{All})$. We follow the phases to generate $G$ until a 'stop' action. By aggregating $\{\mathbf{Adj}_1, \mathbf{Adj}_2, ..., \mathbf{Adj}_B\}$ from the posterior approximation, we estimate the marginal probability of a directed edge from node $i$ to node $j$ as:

$$P_\Omega(i \to j|U_{All}) \approx \frac{1}{B} \sum_{b=1}^B \mathbf{1}(i \to j \in \mathbf{Adj}_b), \tag{47}$$

where $\mathbf{1}(\cdot)$ is the indicator function. This process enables the inference of the underlying interaction graph's structure, which is critical for evaluating the accuracy of SICSM in structural inference tasks. **(d)** For the log-likelihood term, we employ a message-passing neural network to compute future node features:

$$\{\mu_i^{t+1}, \sigma_i^{t+1}\} = \text{MLP}(U_{All_i}, \tilde{Adj}), \tag{48}$$

where $\text{MLP}(\cdot)$ is a neural network that outputs the parameters of the probability distribution for the future state $v_i^{t+1}$:

$$\text{MLP}(U_{All_i}, \tilde{Adj}) = \left\{ U_i^t + f_e \left( \sum_{j \to i \in \tilde{Adj}} f_a(U_i^t, U_j^t) \right) \right\}, \tag{49}$$

where $f_e$ and $f_a$ are multilayer perceptions, and the operation is performed for all nodes. Then we have log-likelihood for each node $i$ at time $t$ can be computed as:

$$\log P(v_i^{t+1}|\lambda, \tilde{Adj}, U_i^t) = \log \mathcal{N}(v_i^{t+1}|\mu_i^{t+1}, \sigma_i^{t+1}). \tag{50}$$

If the node features are multi-dimensional, we set up multiple readout heads in Eqn. 48. This approach effectively handles multi-dimensional node features and incorporates both node features and graph structure, thereby reinforcing the model's predictive accuracy.

## C  More Details about Datasets

In this section, we provide more details about the datasets used in this work apart the description in Section 5.

## C.1 Springs Simulations

To generate these Springs Simulations datasets, we follow the description of the data in [31] but with fixed connections and with 10 nodes, in order to simulate spring-connected particles' motion in a 2D box using the Springs simulation. In this setup, nodes represent particles, and edges correspond to springs governed by Hooke's law. The Springs simulation's dynamics are described by a second-order ordinary differential equation: $m_i \cdot x_i''(t) = \sum_{j \in \mathcal{N}_i} -k \cdot \left( x_i(t) - x_j(t) \right)$. Here, $m_i$ represents particle mass (assumed as 1), $k$ is the fixed spring constant (set to 1), and $\mathcal{N}_i$ is the set of neighboring nodes with directed connections to node $i$, which is sub-sampled from the graphs generated in the StructInfer in previous steps. We integrate this equation to compute $x_i'(t)$ and subsequently $x_i(t)$ for each time step $t$. The resulting values of $x_i'(t)$ and $x_i(t)$ create 4D node features at each time step. To be specific, at the beginning of the data generation for each springs dataset, we randomly generate a ground truth graph and then simulate 12000 trajectories on the same ground truth graph, but with different initial conditions. The rest settings are the same as that mentioned in [31]. We collect the trajectories and randomly group them into three sets for training, validation and testing with the ratio of 8: 2: 2, respectively.

## C.2 NetSims

It is firstly mentioned in [46], which offers simulations of blood-oxygen-level-dependent (BOLD) imaging data in various human brain regions. Nodes in the dataset represent spatial regions of interest from brain atlases or functional tasks. Interaction graphs from the previous section determine connections between these regions. Dynamics are governed by a first-order ODE model: $x_i'(t) = \sigma \cdot \sum_{j \in \mathcal{N}_i} x_j(t) - \sigma \cdot x_i(t) + C \cdot u_i$, where $\sigma$ controls temporal smoothing and neural lag (set to 0.1 based on [46], and $C$ regulates external input interactions (set to zero to minimize external input noise) [46]. 1D node features at each time step are obtained from the sampled $x_i(t)$.

## C.3 Synthetic Biological Networks

The six directed Boolean networks (LI, LL, CY, BF, TF, BF-CV) are the most often observed fragments in many gene regulatory networks, each has 7, 18, 6, 7, 8 and 10 nodes, respectively. Thus by carrying out experiments on these networks, we can acknowledge the performance of the chosen methods on the structural inference of real-world biological networks. We collect the six ground-truth directed Boolean networks from [44] and simulate the single-cell evolving trajectories with BoolODE [44] (`https://github.com/Murali-group/BoolODE`) with default settings mentioned in that paper for every network. We first sample a total number of 12000 raw trajectories. We then sample different numbers of trajectories from raw trajectories and randomly group them into three datasets: for training, for validation, and for testing, with a ratio of $8 : 2 : 2$. After that, we sample different numbers of snapshots according to the requirements of experiments in Section 5.1 with equal time intervals in every trajectory and save them as '.npy' files for data loading.

## C.4 StructInfer Benchmark

The StructInfer benchmark [3] evaluated 12 structural inference methods in a comprehensive way on a synthetic dataset. The dataset covers 11 types of different underlying interaction graphs and two types of dynamical simulations. (`https://structinfer.github.io/`) As there are so many trajectories, we chose the ones under the name 'Vascular Networks', or in short 'VN', whose underlying interaction graphs approximate the real-world vascular networks in biology systems. As the data is already split into three sets: for training, for validation, and for testing, we keep this setting. In the following paragraphs, we describe more details about the Springs and NetSims simulations utilized by the StructInfer benchmark.

For **Springs** simulation, it follows the approach by Kipf et al. [31], to simulate spring-connected particles' motion in a 2D box using the Springs simulation. In this setup, nodes represent particles, and edges correspond to springs governed by Hooke's law. But different from Springs Simulations mentioned above, StructInfer generates ground-truth interaction graphs with the graph properties of the real-world graphs or network. The ground-truth interaction graphs are used to determine the connectivity between the nodes. The Springs simulation's dynamics are described by a second-order ordinary differential equation: $m_i \cdot x_i''(t) = \sum_{j \in \mathcal{N}_i} -k \cdot \left( x_i(t) - x_j(t) \right)$. Here, $m_i$ represents

particle mass (assumed as 1), $k$ is the fixed spring constant (set to 1), and $\mathcal{N}_i$ is the set of neighboring nodes with directed connections to node $i$, which is sub-sampled from the graphs generated in the StructInfer in previous steps. We integrate this equation to compute $x_i'(t)$ and subsequently $x_i(t)$ for each time step $t$. The resulting values of $x_i'(t)$ and $x_i(t)$ create 4D node features at each time step.

For **NetSims** simulation, it is firstly mentioned in NetSim dataset [46], which offers simulations of blood-oxygen-level-dependent (BOLD) imaging data in various human brain regions. Nodes in the dataset represent spatial regions of interest from brain atlases or functional tasks. But different from NeiSim mentioned above, StructInfer generates ground-truth interaction graphs with the graph properties of the real-world graphs or network. The ground-truth interaction graphs are used to determine the connectivity between the nodes. Dynamics are governed by a first-order ODE model: $x_i'(t) = \sigma \cdot \sum_{j \in \mathcal{N}_i} x_j(t) - \sigma \cdot x_i(t) + C \cdot u_i$, where $\sigma$ controls temporal smoothing and neural lag (set to 0.1 based on [46], and $C$ regulates external input interactions (set to zero to minimize external input noise) [46]. 1D node features at each time step are obtained from the sampled $x_i(t)$.

### C.5 PEMS Datasets

These datasets, derived from the California Caltrans Performance Measurement System (PeMS) [11], comprise data aggregated into 5-minute intervals. The adjacency matrix of the nodes is constructed by road network distance with a thresholded Gaussian kernel [48]. Table 1 summarizes these datasets.

Table 1: Statistics of PEMS datasets.

| Dataset | # Nodes | # Edges | # Time Steps | Missing Ratio |
|---------|---------|---------|--------------|---------------|
| PEMS03 | 358 | 547 | $26,208$ | 0.672% |
| PEMS04 | 307 | 340 | $16,992$ | 3.182% |
| PEMS07 | 883 | 866 | $28,224$ | 0.452% |

We resampled the data such that constructing 49 time steps of points for each trajectory, and obtained 12000 trajectories for each with overlapping snapshots. It's important to note that these datasets' adjacency matrices only connect sensors on the same road, omitting alternative connecting paths, which could impact results.

## D   Implementation of Baselines

For the experiments without prior knowledge, we follow the official implementation of the baselines. As for the integrating of the prior knowledge, we leverage different strategies. For the methods based on VAEs, (e.g. NRI, MPM, ACD, iSIDG, RCSI), we directly perform supervised learning on the latent space with known edges, while keep the rest following the original implementation. For JSP-GFN, we set the graph structure in the initial state and reset states the same as prior knowledge.

### D.1   NRI

NRI [31] is a VAE-based model for unsupervised relational inference. We use the official implementation code by the author from `https://github.com/ethanfetaya/NRI` with a customized data loader for our chosen datasets. We add our metric evaluation in the 'test' function, after the calculation of accuracy in the original code.

### D.2   MPM

MPM [12] employs a VAE framework with a relational interaction mechanism and spatio-temporal message passing. We use the official implementation code by the author from `https://github.com/hilbert9221/NRI-MPM` with a customized data loader for our chosen datasets. We add our metric evaluation for AUROC in the 'evaluate()' function of class 'XNRIDECIns' in the original code.

### D.3 ACD

ACD [39] utilizes shared dynamics to infer causal relations within datasets. We follow the official implementation code by the author as the framework for ACD (`https://github.com/loeweX/AmortizedCausalDiscovery`). We run the code with a customized data loader for the datasets in this work. We implement the metric-calculation pipeline in the 'forward_pass_and_eval()' function.

### D.4 ISIDG

iSIDG [51] iteratively refines adjacency matrices to enhance directional inference. We follow the official implementation code by the author as the framework for iSIDG (`https://github.com/wang422003/Benchmarking-Structural-Inference-Methods-for-Interacting-Dynamical-Systems/tree/main/src/models/iSIDG`). We disable the metric evaluations for the AUPRC and Jaccard index in the original implementation of iSIDG for faster computation.

### D.5 RCSI

RCSI [54] integrates reservoir computing for efficient structural inference. We would like to thank the authors of RCSI for the code. Same as iSIDG, we disable the metric evaluations for AUPRC and Jaccard index in the original implementation of iSIDG for faster computation.

### D.6 JSP-GFN

JSP-GFN [18] applies Generative Flow Networks for Bayesian inference of graphical structures. We follow the official implementation code by the author as the framework for JSP-GFN (`https://github.com/tristandeleu/jax-jsp-gfn`). We run the code with a customized data loader for the datasets in this work.

### D.7 SIDEC

SIDEC [53] encodes node dynamics to exploit partial correlations for structural inference. We follow the official implementation code by the author as the framework for SIDEC (`https://github.com/wang422003/SIDEC_torch`). We run the code with a customized data loader for the datasets in this work. By incorporating the prior knowledge, we did not figure out a feasible way to do so. Thus we omit the implementation for integrating prior knowledge.

### D.8 Implementation details of SICSM

The general training pipeline of SICSM is presented in Algorithm 1.

SICSM is implemented with JAX [9], including following packages: dm-haiku [24] and jraph [19]. The implementation of SICSM model consists of two parts: (1) the implementation of Residual Blocks with Mamba, and (2) the implementation of GFN. The implementation of Residual Blocks follows the script of 'mamba-minimal-jax' (`https://github.com/radarFudan/mamba-minimal-jax/tree/main`). We would like to thank the contributors of this repository for all the efforts they have done. And the implementation of GFN of SICSM follows the implementation of JSP-GFN [18], and the authors' implementation can be found at `https://github.com/tristandeleu/jax-jsp-gfn`. We would like to thank the authors for code and hints. The modifications were made to integrate various regularization terms in the graph prior, the log-likelihood, and the data-loading pipelines. Please refer to the link provided in the supplementary document for the exact implementation of SICSM. SICSM is trained with Adam [29] optimizer, with the learning rate as $0.00001$ and for 1000 epochs. Implementation can be found at: `https://github.com/wang422003/SICSM-JAX/`.

Among all, the most important hyperparameter of SICSM would be the number of Residual Blocks in encoder and decoder, $L$ and $L'$, respectively. As we expect a symmetric structure of both, so $L'$ is set as equal to $L$. The exact number of layers actually depends on the number of nodes in the graph, and we report the values of $L$ in Table 2.

**Algorithm 1** The training procedure of SICSM

1: **Input:** trajectory $\mathcal{V}$ of $n$ nodes
2: **Parameters:** number of steps of prefilling $\xi$, learning rate $\alpha$
3: **Parameters:** number of Residual Blocks in encoder $L$, number of Residual Blocks in decoder $L'$
4: **Output:** Structure of the dynamical system $\mathbf{Adj}$
5: Decompose the input trajectory to form present feature set $V^{0:T-2}$ and forecasting feature set $V^{1:T-1}$
6: Initialize the State trajectory at $G_0$ as an unconnected graph with $n$ nodes
7: **repeat**
8:     Get feature-based embedding $\mathbf{h}_i^t$ for every node and for every time step: $\mathbf{h}_i^t = f_{embed}(v_i^t)$
9:     Compose $\mathbf{H}_i = [\mathbf{h}_i^0, \mathbf{h}_i^1, ..., \mathbf{h}_i^{T-2}]$ for every node
10:    Compose $\mathbf{H}_{all} = [\mathbf{H}_i, \text{ for all nodes}]$
11:    **for** $l \leq (L + L')$ **do**
12:       **if** F **then**irst Residual Block
13:          Get the output of the block: $U_0 = f_{RB_1}(\mathbf{H}_{all})$
14:       **else**
15:          Get the output of the block: $U_{RB_l} = f_{RB_l}(U_{RB_{l-1}})$
16:       **end if**
17:    **end for**
18:    **if** C **then**omplete observation of all nodes
19:       Get the embeddings from the last block in encoder: $U_{All} = U_{RB_L}$
20:    **else**
21:       Aggregate all embeddings from the blocks in encoder: $U_{All} = \sum_{l=1}^{L} U_{RB_l}$
22:    **end if**
23:    Project back to input dimension $\hat{V}^{t+1} = f_{proj}(U_{RB_{(L+L')}})$
24:    **for** $\xi$ steps **do**
25:       Sample the stop action probability: $a \sim P_\Omega(\text{stop}|G_k)$
26:       **if** $a$ is the 'stop' action **then**
27:          Reset the State trajectory: $G_{k+1} = G_0$
28:       **else**
29:          Sample $G_{k+1} \sim P_\Omega(G_{k+1}|G_k, \neg\text{stop})$
30:          Store the transition $G_k \rightarrow G_{k+1}$
31:       **end if**
32:    **end for**
33:    Sample $\lambda \sim P_\Omega(\lambda|G, \text{stop})$
34:    Sample $\lambda' \sim P_\Omega(\lambda'|G', \text{stop})$
35:    Evaluate the rewards $R(\langle G, \lambda \rangle)$ and $R(\langle G', \lambda' \rangle)$
36:    Evaluate the loss $\mathcal{L} = \mathcal{L}_{RB} + \mathcal{L}_{GFN}$
37:    Update the parameters of the branch of Residual Blocks and GFN
38: **until** Convergence criterion
39: Sample the approximation of posteriors: $P_\Omega(i \rightarrow j|U_{All}) \approx \frac{1}{B}\sum_{b=1}^{B} \mathbf{1}(i \rightarrow j \in \mathbf{Adj}_b)$
40: Output the sampled $P_\Omega$ as the structure of the investigated dynamical system

Table 2: Number of Residual Blocks in the encoder of SICSM.

| | $n \leq 10$ | $10 < n \leq 30$ | $30 < n \leq 50$ | $50 < n \leq 100$ | $n > 100$ |
|---|---|---|---|---|---|
| $L$ | 5 | 7 | 10 | 14 | 20 |

# E More Experimental Results

## E.1 Supplementary Experimental Results on Other Metrics

## E.2 Experimental Results with Prior Knowledge

The experimental analysis focuses on the impact of integrating varying percentages of prior knowledge (0%, 10%, 20%, and 30%) into the training process of different structural inference models, including NRI, MPM, ACD, iSIDG, RCSI, JSP-GFN, and our proposed SICSM. The results, as depicted in the

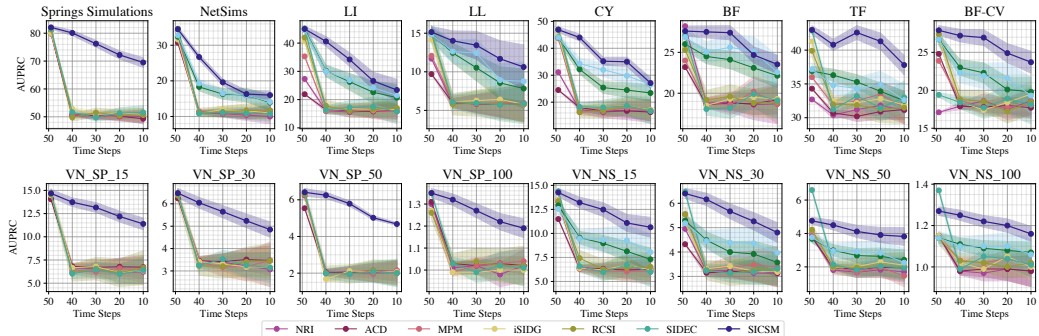

Figure 7: AUPRC values (expressed in percentage) for various methods as a function of the number of irregularly sampled time steps. Results are averaged across ten trials, with time steps varying from 49 to 10. The shadings show the standard deviation of each data point.

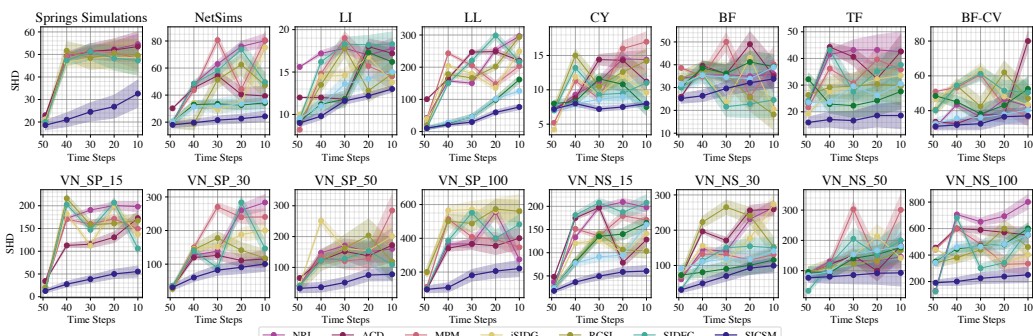

Figure 8: SHD values (expressed in percentage) for various methods as a function of the number of irregularly sampled time steps. Results are averaged across ten trials, with time steps varying from 49 to 10. The shadings show the standard deviation of each data point.

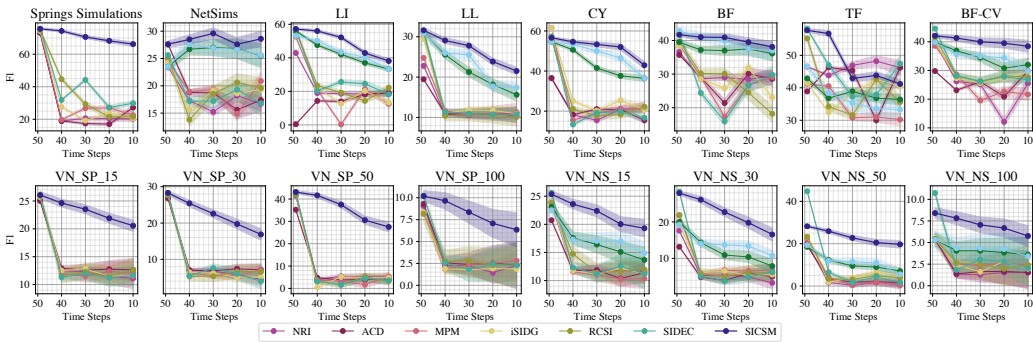

Figure 9: F1 scores (expressed in percentage) for various methods as a function of the number of irregularly sampled time steps. Results are averaged across ten trials, with time steps varying from 49 to 10. The shadings show the standard deviation of each data point.

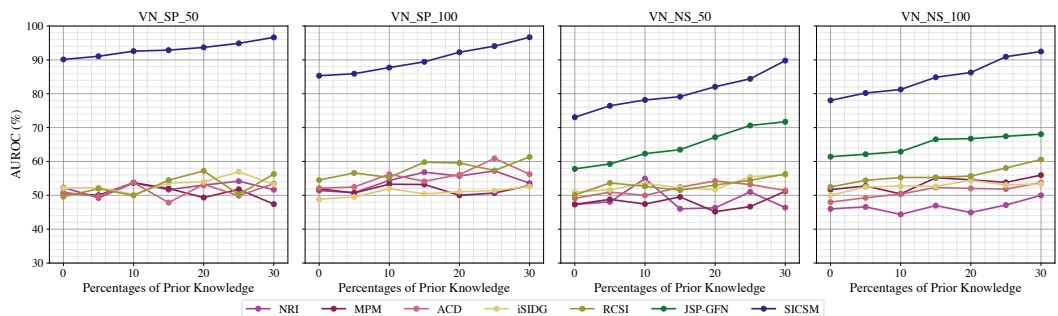

Figure 10: Average AUROC results (in %) of SICSM and baselines with different percentages of prior knowledge on VN datasets.

provided line plots, are evaluated across four datasets: VN_SP_50, VN_SP_100, VN_NS_50, and VN_NS_100. As for SIDEC, we did not figure out a feasible way of integrating prior knowledge into it. The results are shown in Figure 10, and we can see that SICSM consistently outperforms other baseline models across all datasets and percentages of prior knowledge integrated. Notably, SICSM shows a significant improvement in AUROC as the percentage of prior knowledge increases, underscoring its capability to effectively utilize additional information to enhance structural inference accuracy. Models like NRI, MPM, and ACD show moderate improvements with increased prior knowledge but remain less effective compared to SICSM. This suggests that while these models benefit from prior knowledge, their overall adaptability and learning mechanisms might not fully capitalize on the information provided. JSP-GFN and RCSI display variable trends; for instance, JSP-GFN shows notable improvements in the VN_SP_50 and VN_SP_100 datasets but less so in VN_NS datasets, indicating potential dataset-specific sensitivities.

The enhancement in performance with increased prior knowledge is most pronounced in the VN_SP_100 and VN_SP_50 datasets for SICSM. This pattern illustrates the model's robustness in leveraging prior knowledge, particularly in scenarios with larger and possibly more complex network structures. In contrast, the increments in AUROC scores for baselines like iSIDG and RCSI are less steep, suggesting these models, while benefiting from prior knowledge, do not adapt as effectively as SICSM.

To conclude, the integration of prior knowledge markedly benefits the performance of structural inference models, with our SICSM model demonstrating superior capability to utilize such information to enhance prediction accuracy. These results validate the effectiveness of SICSM's design in adapting to additional contextual information, setting a benchmark for future developments in the field. Further investigations could explore optimizing the integration process of prior knowledge to maximize the performance benefits across diverse structural inference scenarios.

### E.3 Experimental Results on PEMS

Table 3: Average AUROC results (%) on PEMS datasets.

|  | PEMS03 | PEMS04 | PEMS07 |
|---|---|---|---|
| JSP-GFN | $60.0_{\pm 1.01}$ | $60.5_{\pm 0.63}$ | $61.2_{\pm 0.70}$ |
| SIDEC | $70.7_{\pm 0.13}$ | $73.5_{\pm 0.18}$ | $70.0_{\pm 0.21}$ |
| SICSM | $71.2_{\pm 0.44}$ | $74.7_{\pm 0.37}$ | $71.2_{\pm 0.47}$ |

This section presents the performance evaluation of two baseline methods, JSP-GFN and SIDEC, alongside our proposed SICSM on three real-world datasets: PEMS03, PEMS04, and PEMS07. These datasets are instrumental in assessing the robustness and effectiveness of structural inference methods in real-world scenarios. The results are summarized in Table 3. Other baselines fail to work on large graphs and encountered OOM errors on these datasets.

As we can see from the table, SICSM consistently exhibits the highest AUROC across all three datasets, with scores of 71.2% on PEMS03, 74.7% on PEMS04, and 71.2% on PEMS07. These results underscore SICSM's superior performance in capturing and predicting complex network

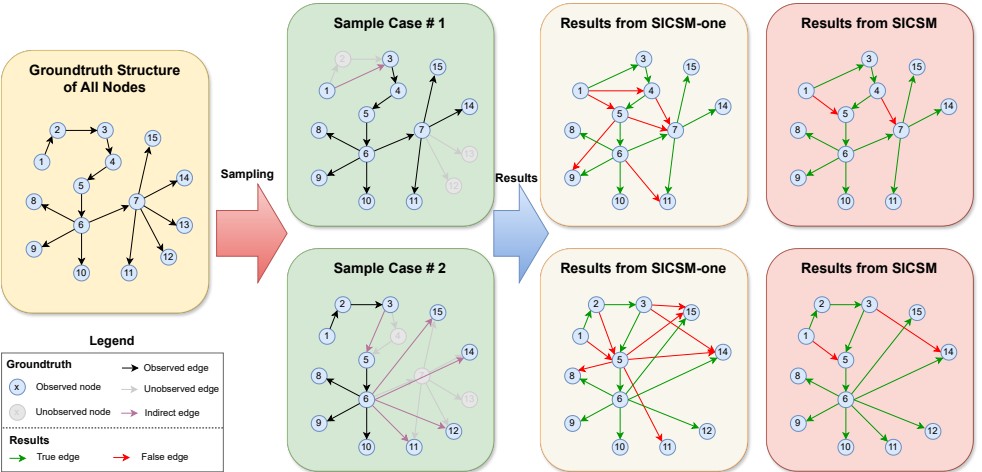

Figure 11: **(1st Column)** Ground truth structure with all nodes. **(2nd Column)** Two examples of 12-node sampling. **(3rd, 4th Columns)** Structural inference results from SICSM-one and SICSM.

dynamics in traffic systems, which is a real-world scenario. SIDEC also performs robustly, especially on the PEMS04 dataset where it achieves an AUROC of 73.5%. However, it slightly trails behind SICSM, particularly on the PEMS03 and PEMS07 datasets. JSP-GFN shows the lowest performance among the evaluated methods, with its highest AUROC at 61.2% on PEMS07, indicating a lesser adaptability to the dynamics of these specific traffic datasets.

The standard deviations reported alongside the AUROC scores indicate the stability of each method's performance across different runs. SICSM demonstrates moderate stability with a standard deviation of approximately 0.44% to 0.47%, suggesting consistent performance despite the inherent variability in real-world data. SIDEC shows the highest stability, particularly on PEMS04, with a minimal standard deviation of 0.18%. This suggests that SIDEC is reliably effective in scenarios represented by this dataset. JSP-GFN, while the least effective in terms of AUROC, maintains a relatively consistent performance as indicated by its standard deviations, which range from 0.63% to 1.01%.

The evaluation on the PEMS datasets validates the effectiveness of SICSM, particularly in comparison to established baseline methods like JSP-GFN and SIDEC. SICSM's ability to consistently outperform other methods underlines its advanced structural inference capabilities, making it a promising solution for complex real-world applications in system dynamics and network analysis.

### E.4 Why Do We Need All Residual Outputs?

As discussed in Section 3.2, in systems with partial observation, it is advantageous to combine learned dynamics from multiple Residual Blocks to enrich the dynamics available for the GFN. However, an intriguing question arises: What is the impact when only the dynamics from the final Residual Block in the encoder are used, similar to approaches used in fully observed systems? This section delves into this query through a detailed case study on a specific dataset. We selected the VN_SP_15 dataset for this examination, focusing on a scenario where 12 nodes (80% of the total) are sampled. We contrasted two configurations of our proposed structural inference method: the comprehensive SICSM, which integrates outputs from all Residual Blocks in the encoder, and a simplified version, SICSM-one, which relies solely on the output from the last Residual Block in the encoder. This comparison aimed to assess the impact of multi-layer output integration on the accuracy and robustness of the inferred network structures, with results illustrated in Figure 11.

Both configurations were evaluated on the same dataset, comprising a full graph and a 12-node sampled version, to appraise their performance across varying degrees of system completeness. Observations from the results indicated that SICSM-one frequently misinterpreted two-hop interactions as three-hop connections, leading to an increased incidence of false positives. This issue arises because SICSM-one relies solely on the output from the final, potentially larger, Residual Block, which can blur the distinctions between one-hop, two-hop, and three-hop dynamics. In contrast, SICSM leverages outputs from multiple layers, enabling a comprehensive representation of dynamics across

Table 4: Average counts of multi-hop negative edges and true positive edges reconstructed upon VN_SP_15 dataset with 12 nodes are sampled with different residual blocks. The average is performed based on 10 runs. For reference, $L = 7$. Each residual block is numbered as their closeness to the input side. For example, Residual Block $[1]$ is the first one, $[1 - 3]$ refers to the integrating outputs from 1 to 3 Residual Blocks.

| Residual Blocks | Count of multi-hop negatives | Count of true positives |
|---|---|---|
| [7] | 7.5 | 10.2 |
| [6] | 7.3 | 9.8 |
| [5] | 6.9 | 9.2 |
| [4] | 6.8 | 8.5 |
| [3] | 6.6 | 8.0 |
| [2] | 6.5 | 7.1 |
| [1] | 6.5 | 7.0 |
| $[1 - 2]$ | 6.5 | 7.2 |
| $[1 - 3]$ | 6.1 | 8.6 |
| $[1 - 4]$ | 5.3 | 9.4 |
| $[1 - 5]$ | 4.2 | 10.0 |
| $[1 - 6]$ | 3.1 | 11.1 |
| $[1 - 7]$ (Full SICSM) | 2.1 | 12.5 |

different temporal dependencies. This multi-layer aggregation is particularly effective at emphasizing shorter connections while still accounting for longer pathways. The integration of shallow blocks plays a crucial role in this configuration, offering detailed insights into shorter dependencies and significantly reducing the likelihood of misidentifying longer connection paths as false positives. Despite these improvements, some inaccuracies remain, suggesting areas for further refinement. Future developments might include implementing an adaptive weighting mechanism that adjusts the influence of dynamics from different Residual Blocks. Such a mechanism would be tailored according to the size of the graph and the longest potential paths within the network, optimizing the model's accuracy in diverse operating conditions.

Moreover, as shown in Table 4, the occurrence of negative multi-hop edges is notably higher when only a single Residual Block is used. This number decreases to 6.5 when only the first block is used, but at the cost of reducing true positive predictions. The best configuration, as highlighted in the table, is the concatenation of outputs from all blocks. This approach not only reduces the occurrence of negative multi-hop edges but also increases the count of true positives, providing a more balanced and accurate representation of the underlying structure.

These findings underscore the effectiveness of multi-layer dynamic integration in SICSM, particularly in settings with partial node observability. They highlight the model's capacity to maintain structural integrity and provide accurate predictions, affirming its potential for broad application in complex, dynamically varying systems.

### E.5 Ablation Study on the Choice of Neural Networks in the Blocks

We conducted additional experiments comparing Transformer [50], LSTM [25], and GRU [15] models on irregularly sampled trajectories with 30 time steps and partial observations with 12 nodes in the VN_SP_15 dataset. The average results from 10 runs are presented in Table 5. For all models, we adjusted the parameters to accommodate the trajectory lengths and performed hyperparameter tuning using Bayesian optimization. As shown in the table, the Transformer model outperforms LSTM and GRU by a small margin, but all are notably inferior to SSSM, as they struggle to effectively handle multi-hop interactions. Additionally, these models perform poorly on irregularly sampled trajectories, as they lack the ability to learn adaptively.

### E.6 Training Time Comparison

The training time analysis for SICSM and baseline methods on the VN_NS datasets, as summarized in Table 6, provides valuable insights into the computational efficiency of these models. The reported times are averaged over ten runs and are presented in hours.

Table 5: Average AUROC results of SICSM with different neural networks in each block. The networks under consideration are Transformer, LSTM and GRU. The experiments are irregularly sampled time steps and partially observed nodes on VN_SP_15 dataset.

| Neural Network | Results on Irre. Sampled | Results on Par. Obser. |
|---|---|---|
| Transformer | 7.5 | 10.2 |
| LSTM | 7.3 | 9.8 |
| GRU | 6.9 | 9.2 |
| SSSM | 6.8 | 8.5 |

Table 6: Training time (hours) of SICSM and baseline methods on VN_NS datasets.

| Methods | VN_NS_15 | VN_NS_30 | VN_NS_50 | VN_NS_100 |
|---|---|---|---|---|
| NRI | 24.6 | 33.5 | 40.5 | 47.1 |
| MPM | 45.3 | 60.4 | 79.2 | 83.6 |
| ACD | 40.2 | 53.1 | 67.6 | 81.7 |
| iSIDG | 43.9 | 56.2 | 88.5 | 98.0 |
| RCSI | 44.6 | 58.0 | 91.6 | 103.4 |
| JSP-GFN | 45.0 | 55.1 | 72.0 | 97.3 |
| SIDEC | 26.8 | 32.9 | 39.5 | 45.0 |
| SICSM | 59.2 | 70.1 | 96.3 | 120.5 |

From the table, it's evident that SICSM, while providing advanced capabilities in structural inference as demonstrated in previous sections, exhibits longer training times compared to both traditional and other state-of-the-art baseline methods. SICSM consistently shows higher training times across all dataset sizes compared to other methods. For instance, at VN_NS_15, SICSM takes approximately 59.2 hours, which is about 35 hours longer than NRI and nearly 33 hours more than SIDEC, the method with the shortest training time for this dataset size. As the size of the dataset increases, the training time for SICSM also increases substantially, from 59.2 hours for VN_NS_15 to 120.5 hours for VN_NS_100. This scaling trend is consistent with other methods but more pronounced in SICSM, suggesting that its complexity scales significantly with larger networks. The reason is that we implement selective SSM with JAX, which lacks the cuda package to boost the selective process that is designed in [20], and the search over all possible state spaces in GFN is time-consuming. The data highlights a crucial area for future development in optimizing the computational efficiency of SICSM. Enhancements might focus on the implementation of JAX-suited cuda package for boosting selective SSM or integrating more efficient learning algorithms to reduce training times without compromising the model's performance.

## E.7 How Good is the Approximation?

In this work, we approximating $P(Adj, \lambda|U_{\text{all}})$ instead of the $Adj$. Thus, it is necessary to evaluate how well $P(Adj, \lambda|U_{\text{all}})$ is approximated with a distributional metric.

Similar to the experiments in JSP-GFN [18], we consider here models over d = 5 variables, with linear Gaussian CPDs. We generate 20 different datasets of N = 100 observations from randomly generated Bayesian Networks. The quality of the joint posterior approximations is evaluated separately for $Adj$ and $\lambda$. For $Adj$, we compare the approximation and the exact posterior on different marginals of interest, also called features in JSP-GFN [18], e.g., the edge feature corresponds to the marginal probability of a specific edge being in the graph. Fig. 12 shows a comparison between the edge features computed with the exact posterior and with SICSM, proving that it can accurately approximate the edge features of the exact posterior. To evaluate the performance of the different methods as an approximation of the posterior over $\lambda$, we also estimate the cross-entropy between the sampling distribution of $\lambda$ given G and the exact posterior $P(\lambda|Adj, U_{all})$. The results are shown in Table 7. We observe that again SICSM samples parameters $\lambda$ that are significantly more probable under the exact posterior compared to other methods.

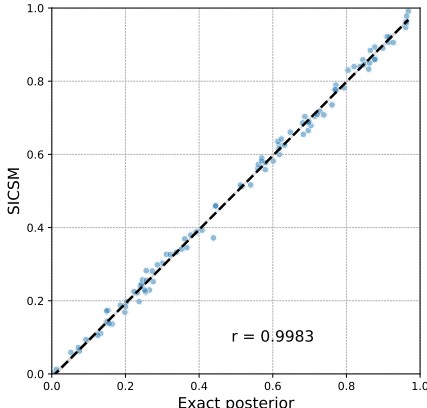

Figure 12: Comparison of the edge features computed with the exact posterior (x-axis) and the approximation given by GFN in SICSM.

Table 7: Comparison with the exact posterior distribution, on small graphs with $n = 5$ nodes. Quantitative evaluation of different methods for joint posterior approximation, both in terms of edge features and cross-entropy of sampling distribution and true posterior $P(\lambda|Adj, U_{all})$. All values correspond to the mean and 95% confidence interval across the 10 experiments.

| | Edge features | | $\mathbb{E}_{Adj,\lambda}[-\log P(\lambda|Adj, U_{all})]$ |
|---|---|---|---|
| | $n \leq 10$ | $10 < n \leq 30$ | |
| JSP-GFN | $0.0019 \pm 0.005$ | $0.998 \pm 0.001$ | $-4.95 \pm 0.51 \times 10^0$ |
| SICSM | $0.0018 \pm 0.007$ | $0.998 \pm 0.001$ | $-4.97 \pm 0.52 \times 10^0$ |

## F   Limitations

While SICSM marks a significant step forward in structural inference, it is imperative to acknowledge its potential limitations for a comprehensive understanding and to guide future research:

- **Reliance on Prior Knowledge Accuracy:** SICSM's enhanced performance through prior knowledge integration is contingent on the accuracy of this information. Misleading or incorrect prior knowledge could adversely impact the model's inference accuracy, leading to potentially flawed conclusions.

- **Prior Knowledge of Edge Existence:** Currently, SICSM leverages prior knowledge about the existence of edges in the graph. However, it is not equipped to incorporate prior knowledge about the non-existence of specific edges, limiting its ability to exclude certain connections during the inference process.

- **Scalability to Very Large Graphs:** The scalability of SIGFN to graphs with an extremely large number of nodes remains untested (e.g., with more than 1,000 nodes). Training and inference in such large-scale graphs may demand significant computational resources and time, which could be a practical constraint. We acknowledged this limitation of SICSM, and currently working on the variant with sub-graph ensemble methods inspired by Cluster-GCN [14] and GraphSAINT [62]. Some methods from federated graph learning may also solve the challenge of scalability of structural inference [26].

- **Evaluation on Synthetic Data:** Due to the challenges in obtaining reliable real-world datasets for structural inference, SICSM has primarily been evaluated on synthetic data in this study. We recognize the potential discrepancies between synthetic and real-world data and plan to address this limitation in future research by exploring real-world applications.

- **Dynamic Graphs Handling:** Currently, SIGFN is formulated for static graphs. However, many real-world graphs are dynamic, with structures that evolve over time. Adapting SICSM to accommodate such dynamic graphs is an essential area for future development.
- **Long Run Time:** As detailed in Appendix E.6, SICSM exhibits the longest running time among all evaluated methods. This extended duration primarily results from the selective SSM in JAX lacking optimized CUDA integration, which is critical for enhancing computational efficiency. Additionally, the time-intensive process of constructing all possible state spaces within the GFN significantly contributes to the overall duration.

Future enhancements to SICSM could involve strategies for validating and correcting prior knowledge, improving scalability and efficiency for handling larger graphs, extending the model's capabilities to dynamic graphs, and implementing efficient selection SSM with JAX as well as boosting the speed of GFN. These advancements will be vital in ensuring SICSM's applicability and reliability across various practical scenarios.

# G   Broader Impact

Much like NRI, MPM, ACD, iSIDG, RCSI, SIDEC, and other structural inference methodologies, SICSM extends its utility to a diverse range of researchers across the realms of physics, chemistry, sociology, and biology, where the uncovering of underlying interaction graph structure is becoming more and more popular. In our investigations, we have demonstrated SISICSM's proficiency in reconstructing graph structures and display robustness to variations in the irregular samplings and incomplete observations, underscoring its versatility and broad applicability. There may be potential societal consequences of our work, none which we feel must be specifically highlighted here.

