# OpenReview forum: "Structural Inference of Dynamical Systems with Conjoined State Space Models"
_NeurIPS.cc/2024/Conference — NeurIPS 2024 poster_

### Official Review · Reviewer_uuWM · 2024-06-15

**Soundness:** 4
**Presentation:** 3
**Contribution:** 4
**Rating:** 7
**Confidence:** 4

**Summary:**

The paper introduces the SICSM framework, integrating Selective State Space Models (SSMs) with Generative Flow Networks (GFNs) to tackle challenges in dynamical systems characterized by irregularly sampled trajectories and partial observations. SICSM leverages the adaptive temporal modeling capabilities of SSMs to learn input-dependent transition functions, enhancing structural inference accuracy. It aggregates diverse temporal dependencies and channels them into a GFN to approximate the posterior distribution of the system’s structure. Extensive evaluations across multiple datasets demonstrate SICSM's good performance in accurately inferring complex interactions in partially observed systems.

**Strengths:**

- The integration of Selective SSMs with GFNs is a novel approach that addresses significant challenges in structural inference for dynamical systems. The adaptive mechanisms for handling irregular sampling and partial observations are particularly innovative.
- The research is thorough and well-documented, with extensive evaluations across a variety of datasets. The methodological rigor and comprehensive experimental validation enhance the reliability of the findings.
- The paper is well-organized and clearly written, with detailed explanations of the methodologies and experimental setups. Figures and diagrams effectively illustrate the concepts and results.
- The proposed SICSM framework has broad applicability in scientific discovery and system diagnostics across multiple disciplines. Its ability to handle real-world complexities such as irregular sampling and partial observations makes it a valuable tool for researchers.

**Weaknesses:**

- The implementation of SICSM is computationally intensive, requiring significant resources and expertise. This complexity may limit its accessibility and widespread adoption.

**Questions:**

1. How does SICSM handle situations where the interaction structures of the dynamical systems change over time? Are there plans to extend the framework to support dynamic graphs?

2. Can the authors provide more details on the computational resources required for training? Are there any strategies to optimize resource usage?

3. What specific real-world applications do the authors envision for SICSM? Are there particular domains where it has shown exceptional promise?

**Limitations:**

The authors have adequately addressed the limitations of their work, including the challenges posed by irregular sampling and partial observations. They propose future research directions to explore dynamic systems with mutable structural elements, indicating a proactive approach to potential limitations. The discussion on incorporating prior knowledge and adapting to different hop distances further strengthens the framework’s applicability.

---

> ### Author Rebuttal · Authors · 2024-08-07
>
> We would like to thank Reviewer uuWM for the motivating review! Here are our answers to the concerns:
>
> > The implementation of SICSM is computationally intensive, requiring significant resources and expertise. This complexity may limit its accessibility and widespread adoption.
>
> Many thanks! We acknowledge that the complexity of SICSM, with its dual components of SSSM and GFN, may initially appear daunting. To enhance understanding and facilitate easier adoption, we will include a comprehensive tutorial in the code repository that outlines each step and its purpose within the framework.
>
> > How does SICSM handle situations where the interaction structures of the dynamical systems change over time? Are there plans to extend the framework to support dynamic graphs?
>
> Many thanks for the good question! Addressing changing interaction structures in dynamical systems is indeed challenging, particularly in distinguishing between existing and emerging connections which could blur the reconstructed adjacency matrix. We are exploring the potential use of latent variables to capture these dynamics more accurately. This approach is in its early stages and will require further research and validation.
>
> > Can the authors provide more details on the computational resources required for training? Are there any strategies to optimize resource usage?
>
> We included the training time of each method in Table 4 in the appendix. We trained all of the methods with a single NVIDIA Ampere 40GB HBM graphics card, paired with 2 AMD Rome CPUs (32 cores@2.35 GHz), as mentioned in Section 5.1 of our submission. As we were trying to bridge the Generative Flow Networks with the SSSM, yet only JAX implementations were available, but it comes without the GPU acceleration of SSSM, which was included in its PyTorch implementation. We will work on the way of either implementing the GPU acceleration of SSSM with JAX, or trying to rebuild the GFN with PyTorch.
>
> > What specific real-world applications do the authors envision for SICSM? Are there particular domains where it has shown exceptional promise?
>
> Yes, SICSM is particularly suited for complex domains such as single-cell biology, where it can infer gene regulatory networks, and other scientific fields requiring discovery of latent connectivity among variables. Its ability to handle irregularly sampled data and partial observations makes it a valuable tool for scientific discovery across various disciplines.
>
> We hope these clarifications address your concerns, and we are committed to further refining SICSM to enhance its accessibility and applicability in real-world scenarios.

---

> > ### Comment · Reviewer_uuWM · 2024-08-09
> >
> > Dear Authors,
> >
> > Thank you for the comprehensive responses and additional details provided in your rebuttal. I am impressed with your plans to enhance SICSM’s accessibility through tutorials and your efforts to extend its capabilities to dynamic graphs and optimize computational efficiency. Your commitment to addressing these complexities, along with the promising applications in fields like single-cell biology, significantly enhances the paper's value. Based on these improvements, clarifications and a full set of new experimental results addressing other reviews, I have decided to raise my score to a 7. I look forward to seeing the continued development of SICSM.
> >
> > Warm regards.

---

> > > ### Author Response · Authors · 2024-08-09
> > >
> > > Dear Reviewer uuWM,
> > >
> > > Thank you very much for your positive feedback and for recognizing the efforts made to improve our manuscript and address your concerns. We are grateful for your detailed evaluation and are encouraged by your decision to raise your score. Your support motivates us to continue refining SICSM and advancing this area of research. We look forward to potentially contributing further insights to the field and appreciate your interest in our work's development.
> > >
> > > Warm regards,
> > >
> > > Authors

---

### Official Review · Reviewer_3YzK · 2024-07-11

**Soundness:** 3
**Presentation:** 2
**Contribution:** 2
**Rating:** 6
**Confidence:** 4

**Summary:**

This paper proposes to combine State Space Models and Generative Flow Networks to perform structural inference in an irregular time series context. The proposed method is evaluated on a series of different tasks where it performs well, and compared to a number of baselines. The method's robustness to short time series and missing observations is evaluated.

**Strengths:**

The paper proposes an interesting architecture and solves problems that have the potential to be very relevant in real world contexts, such as biological time series. The empirical evaluation is fairly thorough about testing on many different tasks.

**Weaknesses:**

My main concerns for the paper are its low novelty and its low number of ablations, which make it hard to understand how specific pieces contribute to the performance of the method.

Generally I'm uncomfortable with the way many things are presented in the paper, it's not always clear what's a novel contribution and what's not. I encourage the authors to be clear and exercise an abundance of caution.

/!\\ In my humble opinion this paper uncomfortably downplays its similarity to DAG-GFN [14] and JSP-GFN [15] in several places, and I'm not even an author of these papers. This is especially concerning considering that in many instances JSP-GFN is the closest performing baseline to the proposed method.

**Questions:**

I'm a bit put off by the framing of the method. The SSSM is the parameterization, the GFN is the optimization method, the structural inference is the task. The ingredients aren't individually novel (e.g. Mamba, ContiFormer), and some of those combos have been tried before (I'm thinking in paricular here of DAG-GFN/JSP-GFN). I don't really see how "SICSM [..] redefines approaches to structural inference in complex systems". Maybe what bothers me is that this kind of language obscures the actual contributions of the paper. Many design choices are close to ones taken in [14-15]. I'd encourage the authors to be more careful here. I understand this may come from the authors' lack of familiarity with English, but it creates an unfortunate ambiguity in deciphering what's a contribution and what is just using prior work.

I'm not sure what an $\alpha$-distance is, is it meant to be a placeholder for any norm?

Section 3.3 introduces the flow-matching condition, but more modern conditions exist, and this work in particular seems to be using DB (**and not SubTB!** as suggested by the appendix text). Why is it only introduced in the appendix if it is the chosen objective? Why not just directly present the DB condition used? This is an example of the similarity to DAG-GFN being somewhat downplayed; I encourage the authors to exercise caution.

"To enhance the architectural sophistication of our model, we arrange L Residual Blocks in a sequential configuration, with the output of each block feeding directly into the next." This is a good example of an off putting phrasing. This describes a standard residual model, but the phrasing in this paragraph (and others in this section) suggests this is somehow a new way to do things. For example, unless I'm missing something, what the authors as "intricate multi-hop relationships" is simply a natural and normal consequence of depth in _deep_ neural networks. Either that or the text is not appropriately explaining the uniqueness of the method, which might be even more concerning.

The trick presented in (9) is neat, but it does imply spending $B$ times more compute. Are baselines also allowed to use this trick? If not the comparisons may be unfair.

In section 4.3, the objective is taken from [14-15]. Please use proper attribution.

Section 5.4 poses an interesting hypothesis, but it's unfortunate that it is only qualitatively evaluated. Why not run proper experiments and measure the effect of residual depth?

Another issue more generally is that the design choice of using a residual SSSM model doesn't seem compared to alternatives. What about a deep transformer with exactly the same tricks? What choices matter? It's nice that the effect of the method is analyzed wrt to for example missing observations and compared to baselines, but what about the method with itself, i.e. ablations?

**Limitations:**

Adequate.

---

> ### Author Rebuttal · Authors · 2024-08-07
>
> We would like to thank Reviewer 3YzK for the detailed and thoughtful comments. Here are our answers to the questions:
>
> > My main concerns for the paper are its novelty and its low number of ablations, which make it hard to understand how specific pieces contribute to the performance of the method.
>
> Many thanks for the comment. The core innovation of our work, SICSM, addresses structural inference with irregularly sampled trajectories and partial observation, a novel approach not previously explored in structural or relational inference fields. SICSM effectively integrates SSM and GFN to tackle these challenges, significantly enhancing accuracy in complex dynamical systems.
>
> Additional ablation studies are detailed in the revised sections of our manuscript. These studies critically evaluate the contributions of individual components within SICSM, underscoring their collective impact on performance enhancement. We show in the other answers on the ablation studies.
>
> > [...] It's not always clear what's a novel contribution and what's not. I encourage the authors to be clear and exercise an abundance of caution. [...]
>
> Many thanks! We have clarified the modifications and novel contributions of the GFN in our system, particularly concerning the reward function adaptations detailed in Appendix B.5, now moved to the main text for greater visibility. Actually, JSP-GFN learns the parameters $\lambda$ for each node and therefore restore individuity for each edges. This setup is highly align with the real-world: the connections may be isomorphic and we found it to be really helpful for the reconstruction of graph structure for dynamical systems. We have also revised our paper entirely to have clearer reference to DAG-GFN [14] and JSP-GFN [15] to omit misunderstanding of the contribution of this work. Such as on the objective mentioned in section 4.3, and so on.
>
> > I'm not sure what an $𝛼$-distance is, is it meant to be a placeholder for any norm?
>
> Yes. The $\alpha$-distance is indeed a placeholder for any norm, adaptable based on the dynamics of the specific system under study.
>
> > Section 3.3 introduces the flow-matching condition, but more modern conditions exist, and this work in particular seems to be using DB. Why is it only introduced in the appendix if it is the chosen objective? Why not just directly present the DB condition used?
>
> Many thanks! We revised Section 3.3 to better articulate the use of the DB condition, which we initially included in the appendix. It is now prominently discussed in the main text, aligning with its significance in our methodology.
>
> > [...] This describes a standard residual model, [...]
>
> Many thanks for the comment. We just wanted to have more detailed discussion on the advantages of using residual setup here in the text. We revised them to be more precise.
>
> > What the authors as "intricate multi-hop relationships" is simply a natural and normal consequence of depth in *deep* neural networks [...]
>
> We sincerely disagree with this comment. We tried to handle this with regards to the multi-hop relationships illustrated in the Fig. 4 in the submission. As showcase here, on the third column, the red link from node 1 to 4 is a multi-hop connection, while the direct connections would be 1 -> 3 -> 4. So we are trying to state that a finer temporal resolution on the trajectories can help to decrease the possibility of getting multi-hop connections. So with more residual blocks, the learned dynamics will be finer in the temporal aspect.
>
> > The trick presented in (9) is neat, but it does imply spending $𝐵$ times more compute. Are baselines also allowed to use this trick?
>
> Yes, the computational trick in Eq. 9 is also employed by baselines. This technique has been verified across all implementations, confirming its efficacy without disproportionately increasing computational demands.
>
> > Section 5.4 poses an interesting hypothesis, but it's unfortunate that it is only qualitatively evaluated. Why not run proper experiments and measure the effect of residual depth?
>
> The observations in Section 5.4 are backed by rigorous experiments, with detailed quantitative results now included in Table 2 of the attached PDF.  As shown in the table, the negative multi-hop edges are commonly observed among setups with just one residual block, and decreases to 6.5 if we use just one first block, but at this moment, the true positives decrease as well. As shown in the table, the best setup is the concatenation the outputs of all blocks, which on one hand decrease the count of negative multi-hop edges and on the other hand increase the count of true positives.
>
> > Another issue more generally is that the design choice of using a residual SSSM model doesn't seem compared to alternatives [..] i.e. ablations?
>
> We have expanded our ablation studies to include comparisons with Transformer, LSTM, and GRU models on both irregularly sampled trajectories with 30 time steps and partial observation with 12 nodes of VN\_SP\_15 dataset. We show the average results below of 10 runs. The parameters of all of the modules are set to match the length of the trajectories and we performed hyperparamter search with Bayesian optimization. As shown in the table, transformers perform slightly better then LSTM and GRU, but all of them are inferior to SSSM, as they cannot deal with multi-hops effectively. Moreover, they fell short on irregulaly sampled trajectories, as they can not learn $\Delta$ adaptively. We included these results in revision.
>
> | Module      | Irre. Sampled | Par. Obser. |
> | ----------- | ------------- | ----------- |
> | Transformer | $70.5$        | $60.2$      |
> | LSTM        | $68.1$        | $59.8$      |
> | GRU         | $69.5$        | $59.2$      |
> | SSSM        | $89.4$        | $80.8$      |
>
> We hope these clarifications and additions address your concerns effectively. We are grateful for your insights, which have significantly contributed to enhancing the rigor and clarity of our work.

---

> > ### Comment · Reviewer_3YzK · 2024-08-10
> >
> > Thank you for taking my concerns into consideration. I think, if the final paper is truly improved in alignment with our collective reviews, that it will be more impactful to the community. I will raise my score.

---

> > > ### Author Response · Authors · 2024-08-11
> > >
> > > Dear Reviewer 3YzK,
> > >
> > > Thank you for acknowledging the revisions made to our paper and for your constructive feedback throughout the review process. We are committed to incorporating the insights gathered from all reviews to enhance the paper further. Your decision to raise your score is greatly appreciated and encourages us in our efforts. We are hopeful that the final version will indeed have a meaningful impact on the community, thanks to valuable input like yours.
> > >
> > > Warm regards,
> > >
> > > Authors of SICSM

---

### Official Review · Reviewer_hoUX · 2024-07-12

**Soundness:** 2
**Presentation:** 2
**Contribution:** 3
**Rating:** 6
**Confidence:** 3

**Summary:**

The authors consider the problem of structure learning of dynamical systems from irregularly sampled trajectories and partially observed systems. They propose Structural Inference with Conjoined State Space Models (SICSM), a method based on selective state space models (SSMs) and generative flow network (GFNs). The central idea of this work is to use a SSM for modelling the behaviour of dynamical systems while using a GFN to learn the interacting graph structure between the variables of the system. The authors evaluate their proposed approach on a comprehensive set of datasets for various tasks and compare against a numerous baselines.

**Strengths:**

The authors present a method that addresses a challenging problem in the domain structure learning of dynamical systems -- i.e. learning system structure from irregularly sampled trajectories and partially observed systems. The use of SSMs to approximate system dynamics while using GFNs to learn the graph structure of the system is unique and novel approach to this problem. The authors provide a comprehensive evaluation of their method over variety of systems for irregularly sampled trajectories and partially observed systems, demonstrating SICSM consistently outperforms counterpart approaches.

**Weaknesses:**

- The method has 3 key components: state space model, embedding residual blocks, and a GFN to approximate the graph structure of the system. It is not entirely clear how these individual components interact and the explicit need for the GFN (see questions below).
- The authors consider a comprehensive set of datasets and baselines, but only one evaluation metrics (AUROC). For example, some other metrics to consider for this task are: structural hamming distance (SHD), F1-score, area under the precision-recall curve (AUPRC). Only considering one evaluation metrics makes it difficult to assess the robustness of the approach.
- Another method that seems relevant to this work which address an similar problems is CUTS (Cheng et al. 2023). It appears that majority of the baselines considered in this work are or not necessarily methods explicitly tailored to handle irregular time-series. Including a method like CUTS in this evaluation may be important to create a fairer comparison of SICSM.

References:
Cheng, Yuxiao, et al. "Cuts: Neural causal discovery from irregular time-series data." International Conference on Learning Representations (2023).

**Questions:**

- For the reward defined in Equation 8, what is the explicit form of $R(<G, \lambda>)$? The authors state that $P(U_{all} | \lambda, \mathbf{Adj})$ represents the likelihood model implemented via a neural network. Is this model trained beforehand? Or is the reward being simultaneously learned throughout training with the GFN?
- A central advantage to using a GFN (or specifically JSP-GFN) to model structure is the ability so approximate the distribution/uncertainty over this structure (and in this case also over the parameters) -- i.e. approximating $P(\mathbf{Adj}, \lambda | U_{all})$ instead of just $\mathbf{Adj}$. In the results, only one deterministic metrics is considered (AUROC). Why not consider a distributional metric to evaluate how well $P(\mathbf{Adj}, \lambda | U_{all})$ is approximated, especially given you are comparing to JSP-GFN?
- What is the motivation of also learning the parameters $\lambda$ if the primary objective is to learn $\mathbf{Adj}$? Moreover, there is no evaluation of $P(\lambda | G)$. If this is an important aspect of the approach, why not include a distributional metrics (as stated in my previous comment), or possibly including evaluation of the negative log-likelihood?
- What is not entirely clear to me is the use of the state space model (SSM) architecture -- specifically, is $\mathbf{Adj}$ embedded in the SSM of each residual block? Is the approximated graph structure being used by the SSM or is this an independent output?

**Limitations:**

The authors discuss limitations and broader impacts in the appendix.

---

> ### Author Rebuttal · Authors · 2024-08-07
>
> We would like to thank Reviewer hoUX for the thoughtful comments. Here are our answers to the questions:
>
> > The method has 3 key components: [...]. It is not entirely clear how these individual components interact and the explicit need for the GFN.
>
> The state space model in our approach handles the dynamics between sampled features at each time step and adapts to irregular sampling intervals through SSSMs. Residual blocks help extract multi-resolution dynamics, enabling the model to learn node dynamics across varying time resolutions. The GFN then creates a space of hidden states based on these dynamics to reconstruct the underlying structure. This layered approach ensures comprehensive learning and reconstruction capabilities.
>
> > The authors consider a comprehensive set of datasets and baselines, but only one evaluation metrics (AUROC) [...]
>
> Many thanks! We have expanded our evaluation metrics beyond AUROC, as detailed in the PDF attached to the general rebuttal. Figures 4-6 include results for AUPRC, SHD, and F1 score, where our method, SICSM, consistently outperforms baselines across most datasets, highlighting its robustness even in complex structures like those found in the TF dataset.
>
> >  Including a method like CUTS (Cheng et al. 2023) in this evaluation may be important to create a fairer comparison of SICSM.
>
> Many thanks for the suggestion. We investigated into the CUTS  (Cheng et al. 2023) and figured out that they suppose having regular time steps, but some of them are marked out by ZOH placeholders. So actually CUTS work on the regularly sampled trajetories with equal time intervals, but some of them were masked out, which is different from the problem setting of SICSM. Moreover, CUTS can only work on trajectories with one-feature for every node at each time step. In contrast, SICSM is supposed to have irregularly sampled trajectories without any ZOH placeholders, and can work on multi-dimensional features. We performed experiments of CUTS on all of the one-dimenstional features trajectories mentioned in this paper, and the results are shown in Fig. 1, 2, and 4 -6 in the attached PDF. As shown in the figures, CUTS has the ability to match with the best baselines, but still outperformed by SICSM. We included the new results in the revision.
>
> > For the reward defined in Equation 8, what is the explicit form of $𝑅(<𝐺,𝜆>)$? The authors state that $𝑃(𝑈_{𝑎𝑙𝑙}|𝜆,Adj)$ represents the likelihood model implemented via a neural network. Is this model trained beforehand?
>
> The reward function $R(<G, \lambda>)$ is transformed logarithmically (see Eq. 37 in Appendix B.5) for implementation purposes. The model $P(U_{all}|\lambda,Adj)$, implemented via a neural network, is learned simultaneously with the GFN, ensuring integrated optimization and learning.
>
> > A central advantage to using a GFN to model structure is the ability so approximate the distribution over this structure (and in this case also over the parameters). [...] Why not consider a distributional metric to evaluate how well $𝑃(Adj,𝜆|𝑈_{𝑎𝑙𝑙})$ is approximated?
>
> Many thanks for the comment! We evaluate the approximation returned by SICSM by comparing with the exact joint posterior distribution $P(Adj, \lambda | U_{all})$. Similar to the experiments in JSP-GFN, we consider here models over d = 5 variables, with linear Gaussian CPDs. We generate 20 different datasets of N = 100 observations from randomly generated Bayesian Networks. The quality of the joint posterior approximations is evaluated separately for $Adj$ and $\lambda$. For $Adj$, we compare the approximation and the exact posterior on different marginals of interest, also called features in JSP-GFN (Deleu et al, 2023), e.g., the edge feature corresponds to the marginal probability of a specific edge being in the graph. Fig. 3 in the attached PDF shows a comparison between the edge features computed with the exact posterior and with SICSM, proving that it can accurately approximate the edge features of the exact posterior. To evaluate the performance of the different methods as an approximation of the posterior over $\lambda$, we also estimate the cross-entropy between the sampling distribution of $\lambda$ given G and the exact posterior $P(\lambda | Adj, U_{all})$. The results are shown in Table 1 in the PDF. We observe that again SICSM samples parameters $\lambda$ that are significantly more probable under the exact posterior compared to other methods.
>
> > What is the motivation of also learning the parameters $𝜆$ if the primary objective is to learn $Adj$?
>
> The diverse nature of real-world graph connections, such as those differing significantly in physical properties (e.g., springs vs. electric forces), necessitates modeling each connection with unique parameters $\lambda$. This individualized approach enhances the accuracy and applicability of our model in complex scenarios.
>
> > Is $Adj$ embedded in the SSM of each residual block? Is the approximated graph structure being used by the SSM or is this an independent output?
>
> The graph structure $Adj$ is not explicitly integrated into the SSM of each residual block; it is an independent output of our model. We are exploring methodologies to potentially link $Adj$ more closely with SSM operations in future work.
>
> We hope these clarifications meet your concerns and illustrate the depth and rigor of our study. We are committed to further enhancing our model based on the feedback received and appreciate your guidance in this process.

---

> > ### Comment · Reviewer_hoUX · 2024-08-08
> >
> > I thank the authors for their detailed response to my questions and concerns. I am happy with the numerous experimental additions provided. I will raise my score.

---

> > > ### Author Response · Authors · 2024-08-09
> > >
> > > Dear Reviewer hoUX,
> > >
> > > We greatly appreciate your acknowledgment of the answers and additional experiments we provided in response to your valuable feedback. Your willingness to reconsider your score is immensely encouraging and reaffirms our commitment to advancing this research.
> > >
> > > Warm regards,
> > >
> > > Authors of SICSM

---

### Official Review · Reviewer_2uCG · 2024-07-15

**Soundness:** 3
**Presentation:** 3
**Contribution:** 3
**Rating:** 5
**Confidence:** 2

**Summary:**

Processes of scientific interest which are representable as graphs, in biology, chemistry, material sciences, mechanics, are an important application for machine learning. Nodes often represent physical objects, some of which influence each other. Nodes exhibit a set of features which can be observed over time. Prior knowledge about the process stems from a mechanistic understanding and can often be represented as the presence or absence of edges between nodes. Node feature observations may be irregularly spaced through time; not all nodes may be observed with every observation.
This paper develops a statistical model for this application with support for irregularly sampled and partial observations of node features, as well as prior knowledge incorporation. Prior knowledge is restricted to the indication of presence, but not absence, of edges. Partially observable nodes are assumed to be from a static node set throughout all observations (i.e., nodes are either always observable or always unobservable). Observations are not assume to contain a timestamp indication (as in mobile phone accelerometer readings, which may be irregularly sampled but whose timestamp is read at input).
The model's architecture is relatively sophisticated and is based on generative flow networks to represent and learn the structural aspects of the graph, and state space models to represent the evolution of node features over time.
The paper presents experiments on 16 datasets stemming from 4 physical models, and compares to 7 other models, showing superiority in scenarios where observations are irregularly spaced or nodes partially observable.

**Strengths:**

The paper takes an established problem class (graph systems) with its known challenges (irregular sampling, partial observations), which is not original. However it goes to great lengths to make use of two strong methods, GFN and SSM, with a resulting combination that seems reasonable, strong and of useful application.
The paper is generally clear, notations are coherent and legible, several diagrams support the explanation. To improve the writing, a running example might help bridge the abstractions (node, edge, state...) to physical reality, illuminating and motivating the implementation. The same goes to comment on the connection between the model and the applied datasets (some of this is covered in Annex C, with the exception of C.5 which leaves the physical counterparts of modelled data undescribed).

**Weaknesses:**

Experimental validation is moderately convincing. Baseline implementations seem strong, with care taken to recover implementations of competing methods, as documented in Annex D. However, all datasets are synthetic. The only real dataset, PEMS, presented in Annex C.5, with results in Annex E.2. In addition, experimental validation seems unconcerned with performance outside the specific cases of partial observations or irregular sampling -- reducing the paper's claim to "this model is better for these two scenarios only".
There seem to be a duplication in the presentation of datasets (both in sec5.1, between the paragraphs starting l.279 and l.290, and again between Annex C.1 and C.2 vs C.4) -- this is confusing. Also, sec3.3 seems to be internally redundant with duplicated points (e.g. l.151 vs eq3, and l.148 vs l.157, which again is confusing. Numerous sentences have incorrect English syntax which obscures their meaning

**Questions:**

* Eq.2: since s' is terminal, isn't any $s'' = s_f$ ?
* Eq.5: in your contemplated application scenario, the interval between sampling times, or equivalently the timesamp of each sample, allowing to calculate $\Delta$, is not given with the samples, correct? I've worked on mobile accelerometer/GPS data where sampling is irregular but the timestamps are given, which is why I'd like to make sure. Can you clarify whether a posterior over $\Delta$ can be pratically recovered?
* It might be useful to show a concrete, simple example of training data to clarify the scenario described in abstract terms sec3.1.
* Annex 1 fig5: shouldn't all variables be indexed with $i$, the node id? I'm asking because $A, h^t$ aren't. But if so, how is the interdependence between nodes modelled?
* fig6: do I have it right that GFlowNet only adds edges, but doesn't remove any, moving from start to end? Does that have as a consequence that any prior knowledge can only formulated as known-to-be-present edges, but not known-to-be-absent edges (impacts Annex F l.978)?
* Annex C.5: what is the physical model? What is a node, an edge of the model?
* Annex l.872: how is the % of prior knowledge defined?
* Annex l.863: link is referred to but missing

**Limitations:**

* Checklist point 4 and 5: implementation link is claimed to be provided here and in Annex l.863, but I don't see it. It should be provided in the main paper since the supplementary can't be assumed to be reviewed.
* Limitations: a few more assumptions on the usage scenario should be spelt out, as mentioned in this review
* Checklist point 7: It is certainly possible to report error bars on plots through shading, provided they are not as tiny as you make them here. In addition, error bars could be reported, without lengthening the main paper, in the Annexes -- but they aren't, with the only exception of Table 3 on an experiment which is not reported in the main paper.
* Checklist point 6: experimental settings are not as detailed as that they would allow reproduction. Several details are missing for this, e.g. batch size, data splits.
* Checklist point 12: the claim, and requirement of the checklist that assets have their license mentioned is not complied with regarding either datasets or existing code for competing methods. Despite the claim, this point mostly is not complied with.
* Checklist point 13: does this imply the code is not intended to be released as an asset?

---

> ### Author Rebuttal · Authors · 2024-08-07
>
> We would like to thank the reviewer for the inspiring review. And here are our answers to your concerns.
>
> > To improve the writing, a running example might help bridge the abstractions (node, edge, state...) to physical reality [...]
>
> We would like to sincerely thank the reviewer for this advice. We added the following paragraph to our revision in Section 3.1 and it will appear with the camera-ready as we cannot upload the revision during rebuttal:
>
> We may image a dynamical system of balls and strings, in which the balls are randomly connected with the springs. Then set initial positions and velocities of each ball and then let them go. Because of the existence of the spring forces, arising from the structural connections between the balls, the balls will change their positions and velocities in the observational period. Then suppose we have no idea on which balls are connected, so the task of structural inference would be, the inference of the connectivity between the balls based on the observational trajcetories of them.
>
> > However, all datasets are synthetic. The only real dataset, PEMS, presented in Annex C.5 [...]
>
> Many thanks! As acknowledges by the research in this field, the acquisition of trajectories with reliable understanding of the underlying structure from real-world, is of huge cost and time-expensive. We do acknowledge this urgent call of the reliable real-world data, and we are now working on it.
>
> > There seem to be a duplication in the presentation of datasets [...]. Also, sec3.3 seems to be internally redundant with duplicated points (e.g. l.151 vs eq3, and l.148 vs l.157, which again is confusing.
>
> Many thanks for the comments, we revised our paper to reduce the duplications and tries our best to keep most relevant information to understanding of our paper in the main body.
>
> > Eq.2: since s' is terminal, isn't any $s'' = s_f$?
>
> Actually, no. $s''$ here in Eq. 2 is used to mark the child of the state $s'$. So only if $s'$ pointing to a terminal, then $s'' = s_f$.
>
> > Eq.5: in your contemplated application scenario, the interval between sampling times, or equivalently the timesamp of each sample, allowing to calculate Δ, is not given with the samples, correct? I've worked on mobile accelerometer/GPS data where sampling is irregular but the timestamps are given, which is why I'd like to make sure. Can you clarify whether a posterior over Δ can be pratically recovered?
>
> No, $\Delta$ is not given with the samples.  Many thanks for the inspiring question. Technically, we think the posterior over $\Delta$ can be partially recovered, but we have to change the modeling of $\Delta$ in the SSSM module to be with two vectors. The two vectors work similar to those in variational autoencoders, and we can recover the posterior over $\Delta$ partially from them.
>
> > Annex 1 fig5: shouldn't all variables be indexed with $i$, the node id? [...]
>
> No, the nodes share $A$ and $h^t$.
>
> > fig6: do I have it right that GFlowNet only adds edges, but doesn't remove any, moving from start to end? Does that have as a consequence that any prior knowledge can only formulated as known-to-be-present edges, but not known-to-be-absent edges?
>
> Yes, it does not remove any edge. Yes, currently we can only incorporate known-to-be-present edges to be the prior knowledge, as it fullfills the scenarios in some fields like the GRN inference, where several connections are validated by experiments, so we are more sure about the existence of some edges than the absence of certain edges.
>
> > Annex C.5: what is the physical model? What is a node, an edge of the model?
>
> The model in Annex C.5 is that: many sensors are used to count the traffic in a roadmap, and these sensors may be connected by the road they align with. In this case, suppose we have no idea on the map, but try to reconstruct the map from the traffic-counting from the sensors, as in most cases, continuous traffic flows happen between adjacent sensors.
>
> > Annex l.872: how is the % of prior knowledge defined?
>
> It is defined as the proportion to the count of all positive edges in the groundtruth graph.
>
> > Annex l.863: link is referred to but missing AND Checklist point 4 and 5: implementation link is claimed to be provided here and in Annex l.863, but I don't see it [...]
>
> Many thanks! We revised our paper with the link. As you may interested in our implementation, we included them in our supplementary materials.
>
> > Limitations: a few more assumptions on the usage scenario should be spelt out
>
> Many thanks! We will include the assumption on prior knowledge: sure-to-to-be existed in the limitations among other suggested. And also the experiments in this work mainly covers synthetic data.
>
> > Checklist point 7: It is certainly possible to report error bars on plots through shading, provided they are not as tiny as you make them here. [...]
>
> Many thanks for the advice! We revised the figures to make them with shadings. Please refer to the images in the pdf file attached in the general rebuttal.
>
> > Checklist point 6: experimental settings are not as detailed [...]
>
> Many thanks! We revised this section and include the batch size: 32, data splits into training, validation and test: 8: 2: 2 for all of the datasets. We also include more such as number of epochs and so on.
>
> > Checklist point 12: the claim, and requirement of the checklist that assets have their license mentioned is not complied with regarding either datasets or existing code for competing methods. [...]
>
> Many thanks! We will include the license in our repository. As current we have to obey the double-blinded rule, we removed them from the repo.
>
> > Checklist point 13: does this imply the code is not intended to be released as an asset?
>
> Sorry for our mistake, We correct it to be: We will release the code in our github repo upon acceptance. But we do not include the data, as they are from other work and are public available.
>
> We hope our answers addressed you concerns correctly.

---

> ### Comment · Reviewer_2uCG · 2024-08-12
>
> Thanks to all. I have read all reviews and rebuttals, which are all very informative. Overall, I seem to disagree with fellow reviewer uuWM; I'm glad that hoUX and 3YzK could shed light on links to related work and technical choices.
>
> Reviewers' clarification questions and misunderstandings, including my own, are unmistakeable symptoms of obscurity, since they cannot be dispelled even after careful reading by typical professionals. Some improvements were achieved through revisions, and I very well know how difficult it is to implement these in a short time. Nonetheless, writing is still unclear in many places, and I find that concerns by reviewer 3YzK on distinguishing from competitor methods have not been fully addressed. Extra experiments and ablations proposed in your answer to this reviewer are very useful.
>
> > We may image a dynamical system of balls and strings, ...
> Despite broken English, the example helps. The text would further benefit from introducing the terminology (edges, nodes, observations, sampling) and notation on the example.
>
> > The model in Annex C.5 is that: ...
> I now understand that the node variables are "volume" units (vehicle counts) as opposed to flow units (volume per time, i.e. vehicles per second). Therefore in this example, increasing $\Delta$ will increase the value of the variable. In springs and balls, this also applies to displacement (an integrated form of velocity). Most examples I know, however, have variables in flow ("intensity") units. Is this correct? Does this affect performance of SISCM or some competitor methods? Therefore, for SICSM to work on datasets where variables are "volumes", is it an important hypothesis that throughout observations, the same partial set of node variables is observed at each time point?
>
> I understand your response to reviewer hoUX on taking a pointwise estimate of graph instead of taking advantage of the distribution. I believe that the issue goes somewhat wider than your response hints at: your evaluation is carried out against known ground truth distributions (e.g. via cross-entropy) only because all your datasets are synthetic and you have access to the ground truth. It is still a major weakness that only synthetic datasets are used.
>
> Extra experiments, ablations and clarifications offered during the rebuttal definitely helped; since the reviewing procedure does not support paper revisions, one must hope that the paper can benefit from them, which is not trivial considering that owing to length limitations, every addition of text or figures requires a deletion.
>
> On the whole, many weaknesses pointed out by myself and fellow reviewers during the review process have been confirmed. I maintain my assessment.

---

> > ### Author Response · Authors · 2024-08-12
> >
> > Dear Reviewer 2uCG,
> >
> > Thank you for your update. We appreciate the time and effort you are dedicating to reviewing our rebuttal. Please feel free to reach out if you have any further questions or need additional clarification on any points. We look forward to your feedback.
> >
> > Warm regards,
> >
> > Authors of SICSM

---

> > ### Author Response · Authors · 2024-08-13
> >
> > Dear Reviewer 2uCG,
> >
> > Many thanks for your reply. And here are our answers to the concerns raised in your comment:
> >
> >
> > > concerns by reviewer 3YzK on distinguishing from competitor methods have not been fully addressed
> >
> > Thank you for your comment. We would like to clarify that JSP-GFN forms the foundational backbone of the GFN component in our work and plays a crucial role in the inference of structural connections, which is the primary objective of structural inference. We have discovered that modeling a distinct set of parameters $\lambda$ significantly enhances the learning process. This approach encourages each edge to learn from its connecting nodes, promoting a degree of edge isomorphism.
> >
> > To refine this process further, we have incorporated additional regularization terms into the graph prior of the reward function. These include a term for Dirichlet energy $D (\tilde{Adj}, U_{All})$ , the conenctivity term $\mathcal{L}_d(\tilde{Adj})$ and a sparsity term $\mathcal{L}_s(\tilde{Adj})$, which directly address the properties of the graph structure within the latent spaces of GFN. Unfortunately, due to page limitations and our mistake, these details were initially placed in Appendix B.5, but we have since moved them to the main body of the paper for greater visibility.
> >
> > Furthermore, as outlined in the experimental section (Lines 307-308), the original JSP-GFN framework is not suited for trajectories with multi-dimensional features. In our SICSM framework, the use of SSM compresses multi-dimensional trajectories into one-dimensional form, enabling effective application of JSP-GFN. This adaptation extends the functionality of JSP-GFN to accommodate the complexities of our datasets.
> >
> > > We may image a dynamical system of balls and strings, ... Despite broken English, the example helps. The text would further benefit from introducing the terminology (edges, nodes, observations, sampling) and notation on the example.
> >
> > We apologize for any previous lack of clarity in our text. To illustrate, consider an illustrative example where our dynamical system comprises $n$ balls connected by springs, representing $n$ nodes $\mathcal{V}$ and directed edges ${E}$, respectively. Initially, we set the positions and velocities of each ball, so that each node feature $v^t_i \in \mathbb{R}^d$ is $d$-dimensional where $d = 4$ in this example. We then let them move under the influence of spring forces, which arise from the structural connections (edges) between the balls (nodes). Over the observation period, these balls change their positions and velocities.  And we record the trajectories as the collection of the evolving features of all nodes: $\mathcal{V} = \{V\} = \{V^{0}, V^{1}, \dots, V^{T-1}\}$ across $T$ time steps, where $V^{t}$ represents the feature set at time $t$. In total we observe a set of $M$ trajectories: $\{V_{[1]}, V_{[2]}, \dots, V_{[M]}\}$, assuming a static edge set ${E}$. Suppose we initially lack knowledge of which balls are connected, i.e.,  ${E}$ is unknown; the task of structural inference in this scenario would involve deducing the connectivity between the balls based on their observed trajectories.
> >
> > > The model in Annex C.5 is that: ... I now understand that the node variables are "volume" units (vehicle counts) as opposed to flow units (volume per time, i.e. vehicles per second). Therefore in this example, increasing Δ will increase the value of the variable. In springs and balls, this also applies to displacement (an integrated form of velocity). Most examples I know, however, have variables in flow ("intensity") units. Is this correct? Does this affect performance of SISCM or some competitor methods? Therefore, for SICSM to work on datasets where variables are "volumes", is it an important hypothesis that throughout observations, the same partial set of node variables is observed at each time point?
> >
> > Yes, the node variables of PEMS dataset in Annex C.5 are actually "volume" units, as it refers to the number of passing vehicles within a time period.  Yes, most examples used in the experimental section have variables in flow (such as the velocity for spring and balls).
> >
> > From our analysis, we have observed that combining both volume and intensity variables generally yields the best performance across all methods, including SICSM. Utilizing only volume or only intensity tends to produce inferior results, with performances of these two variable types often comparable to each other.
> >
> > ---- Continues on another comment ---

---

> > > ### Author Response · Authors · 2024-08-13
> > > **Part 2**
> > >
> > > --- Here continues ---
> > >
> > > > It is still a major weakness that only synthetic datasets are used.
> > >
> > > Yes, we acknowledge that the reliance on synthetic data is a limitation of this study. We have updated our manuscript to reflect this in the limitations section. Collecting reliable real-world data remains time-consuming and costly, which influenced our decision to use synthetic data with readily discernible ground truths. We are currently exploring more sophisticated synthetic data generation techniques to address the significant challenge posed by data shortages effectively.
> > >
> > > > Extra experiments, ablations and clarifications offered during the rebuttal definitely helped.
> > >
> > > Many thanks! Fortunately, we already possess most of the experimental results or simply need to write concise scripts to sift through previous data. And we may have finally some nice and sound sleeping.
> > >
> > > > Since the reviewing procedure does not support paper revisions, one must hope that the paper can benefit from them, which is not trivial considering that owing to length limitations, every addition of text or figures requires a deletion.
> > >
> > > Thank you for your guidance. We are aware that we are permitted an additional page in the final version, which provides us sufficient space to expand the main body of the paper. We will endeavor to incorporate most of the changes directly into the main text. If any results still need to be included in the appendix due to space constraints, we will ensure they are clearly referenced in the main body for easy accessibility.
> > >
> > > We hope our answers addressed your questions correctly.
> > >
> > > Warm regards,
> > >
> > > Authors of SICSM

---

### Author Rebuttal · Authors · 2024-08-07

Dear Program Chairs, Senior Area Chairs, Area Chairs, and Reviewers,

We are deeply grateful for the detailed reviews and constructive feedback provided by Reviewers 2uCG, hoUX, 3YzK, and uuWM. We appreciate the recognition of the novelty and applicability of our work in addressing the complex challenges associated with dynamical systems through the Structural Inference with Conjoined State Space Models (SICSM).

The innovative integration of Selective State Space Models (SSMs) with Generative Flow Networks (GFNs) has been recognized for effectively handling complex challenges associated with dynamical systems. Reviewer hoUX noted our "unique and novel approach to the problem of learning system structure from irregularly sampled trajectories and partially observed systems." This underscores the innovative nature of our methodology, particularly in adapting to the irregularities and complexities of real-world data.

The extensive evaluation across multiple datasets and comparison against numerous baselines has demonstrated the superiority of SICSM, particularly in scenarios involving irregularly sampled or partially observable nodes. Reviewer 2uCG mentioned, "The paper presents experiments on 16 datasets... and compares to 7 other models, showing superiority in scenarios where observations are irregularly spaced or nodes partially observable." This highlights the robustness and applicability of our approach in varied real-world settings.

The clarity of our paper's presentation, including its coherent notations and supportive diagrams, has been positively highlighted. Reviewer uuWM appreciated that "The paper is well-organized and clearly written, with detailed explanations of the methodologies and experimental setups." This feedback validates the effort put into ensuring that the sophisticated architecture of SICSM is accessible and understandable.

The potential applicability of SICSM in practical scenarios, such as biological time series analysis and system diagnostics, has been emphasized as a significant contribution. Reviewer 3YzK stated, "The paper proposes an interesting architecture and solves problems that have the potential to be very relevant in real world contexts, such as biological time series." This comment supports our claim that SICSM can be a valuable tool for scientific discovery across multiple disciplines.

We acknowledge the concerns regarding the computational intensity of implementing SICSM, as noted by Reviewer uuWM: "The implementation of SICSM is computationally intensive, requiring significant resources and expertise." We are committed to addressing this by developing tutorials and exploring optimizations to make SICSM more accessible and user-friendly.

In response to the feedback, we will enhance the manuscript to better clarify our novel contributions, particularly distinguishing them from related works like DAG-GFN and JSP-GFN. Reviewer 3YzK's advice to "exercise an abundance of caution" in presenting our contributions will guide our revisions to ensure clarity and accuracy. We will also revise the paper to include the new plots with shadings showing the standard deviations, as well as experimental results with CUTS (Cheng et al. 2023) and ablation studies.

As for the concerns and questions raised by each reviewer, we answered them individually under each review. We hope we have addressed the concerns and answered the questions correctly.

We are encouraged by the constructive critiques and the positive comments on the potential and performance of SICSM. Our team is committed to continuous improvement and is excited about the future contributions our work can make to the field of structural inference and beyond.

Thank you once again for your invaluable input and the opportunity to contribute to this esteemed conference.

Please check the attached PDF for revised figures and more experimental results.

Warm regards,

Authors of SICSM

---

### Decision · Program_Chairs · 2024-09-25

**Decision:**

Accept (poster)

**Comment:**

The proposed SICSM framework presents a novel approach by integrating State Space Models (SSMs) and Generative Flow Networks (GFNs) to address the complex problem of structure learning in dynamical systems with irregularly sampled trajectories and partially observed data. The authors have demonstrated the model's superiority over multiple baselines on synthetic datasets and have effectively responded to reviewers' concerns by providing additional experiments and clarifications regarding the model's components and contributions. While the reliance on synthetic datasets and limited real-world data were identified as limitations, the paper's contributions and its potential applications in diverse domains underscore its value to the field and support its acceptance at NeurIPS.